# Biomimetic computer-to-brain communication enhancing naturalistic touch sensations via peripheral nerve stimulation

Giacomo Valle [1,12], Natalija Katic Secerovic [1,2,3,12], Dominic Eggemann[1], Oleg Gorskii[4,5,6], Natalia Pavlova[4], Francesco M. Petrini[7], Paul Cvancara [8], Thomas Stieglitz [8], Pavel Musienko[4,9,10], Marko Bumbasirevic[11] & Stanisa Raspopovic [1] ✉

Artificial communication with the brain through peripheral nerve stimulation shows promising results in individuals with sensorimotor deficits. However, these efforts lack an intuitive and natural sensory experience. In this study, we design and test a biomimetic neurostimulation framework inspired by nature, capable of "writing" physiologically plausible information back into the peripheral nervous system. Starting from an in-silico model of mechanoreceptors, we develop biomimetic stimulation policies. We then experimentally assess them alongside mechanical touch and common linear neuromodulations. Neural responses resulting from biomimetic neuromodulation are consistently transmitted towards dorsal root ganglion and spinal cord of cats, and their spatio-temporal neural dynamics resemble those naturally induced. We implement these paradigms within the bionic device and test it with patients (ClinicalTrials.gov identifier NCT03350061). He we report that biomimetic neurostimulation improves mobility (primary outcome) and reduces mental effort (secondary outcome) compared to traditional approaches. The outcomes of this neuroscience-driven technology, inspired by the human body, may serve as a model for advancing assistive neurotechnologies.

Loss of communication between the brain and the rest of the body due to an injury or a neurological disease severely impacts the sensorimotor abilities of disabled individuals. Often, they also experience the inability to sense their own body. The resulting low mobility and accompanying loss of independence cause severe health problems and necessitate continuous care. Recently developed neurotechnologies[1-3] exploit direct electrical stimulation of the residual peripheral or central nervous system to restore some of the lost sensorimotor functions.

[1]Laboratory for Neuroengineering, Department of Health Sciences and Technology, Institute for Robotics and Intelligent Systems, ETH Zürich, Zürich, Switzerland. [2]School of Electrical Engineering, University of Belgrade, 11000 Belgrade, Serbia. [3]The Mihajlo Pupin Institute, University of Belgrade, 11000 Belgrade, Serbia. [4]Laboratory for Neuroprosthetics, Institute of Translational Biomedicine, Saint-Petersburg State University, Saint-Petersburg, Russia. [5]Laboratory for Neuromodulation, Pavlov Institute of Physiology, Russian Academy of Sciences, Saint Petersburg 199034, Russia. [6]Center for Biomedical Engineering, National University of Science and Technology "MISIS", 119049 Moscow, Russia. [7]SensArs Neuroprosthetics, Saint-Sulpice CH-1025, Switzerland. [8]Laboratory for Biomedical Microtechnology, Department of Microsystems Engineering–IMTEK, Bernstein Center, BrainLinks-BrainTools Center of Excellence, University of Freiburg, D-79110 Freiburg, Germany. [9]Sirius University of Science and Technology, Neuroscience Program, Sirius, Russia. [10]Laboratory for Neurorehabilitation Technologies, Life Improvement by Future Technologies Center "LIFT", Moscow, Russia. [11]Orthopaedic Surgery Department, School of Medicine, University of Belgrade, 11000 Belgrade, Serbia. [12]These authors contributed equally: Giacomo Valle, Natalija Katic Secerovic. ✉e-mail: nesta.fale@gmail.com

Indeed, brain-computer interfaces (BCIs) exploiting implantable neural devices could potentially restore the bidirectional flow of information from and to the brain[1,4,5]. The implant of bio-compatible electrodes in the residual neural structures[6] allows one to create a direct communication channel with the brain. Indeed, neural stimulation of the peripheral somatic nerves (PNS)[7–10], spinal cord[11–15], and somatosensory cortex (S1)[16–18] showed the ability to restore missing sensations, resulting in closed-loop neuroprostheses able to establish a bidirectional link between humans and machines. Sensory feedback restoration improved patients' ability to use bionic limbs and increased prosthesis acceptance rate[5,19–22]. However, the resulting dexterity of bionic hands is still far from that of natural hands in able-bodied individuals[23], and mobility and endurance achieved with bionic legs are also to be improved[24]. This is due to multiple factors, among which current neurotechnologies fail to induce natural sensation[1] and often result in unpleasant paresthesia instead. Indeed, common neuromodulation devices do not stimulate neurons based on the human natural touch coding or using model-based approaches[25–27], but rather with predefined constant stimulation frequency[28–30]. These stimulation patterns activate all neurons simultaneously, contrary to neural activity during in-vivo natural touch[31]. In fact, this natural asynchronous activation is driven by the probabilistic nature of action potential generation in sensory organs, such as muscle spindles[32] or touch afferents[33], and by the stochastic nature of synaptic transmission[34]. Synchronized stimulation, which generates an unnatural aggregate activity within the neural tissue, could be among the causes of paresthesia percepts[8,26]. In fact, paresthetic sensations are likely to arise from this unnatural fiber activation[35] and can be due to the over-excitation of afferents or a cross-talk between them[36]. When caused by neuropathies, paresthesia is often chronic and does not improve over time, which might reflect an inability of the central nervous system to learn how to interpret such aberrant neural responses[31], making the use of electrical stimulation challenging. Moreover, it can interfere with the individual's ability to sense and respond to other types of sensory information, such as touch or temperature. This can make it difficult to perform certain tasks or activities that require the use of multiple senses or to interact with objects in the environment.

As a possible answer to this problem, the electrical stimulation built by mimicking the natural tactile signal (known as biomimetic sensory feedback[31,37]) has been shown to evoke more intuitive and natural sensations that better support interactions with objects, compared to commonly used stimulation paradigms[38–40]. These biomimetic approaches may have the ability to electrically evoke aggregate population response similar to the natural ones[23,41]. Previous studies on natural touch suggest that somatosensory information about most tactile features is encoded synergistically by all afferent classes in the nerve[42]. Importantly, the somatosensory cortex[43,44] (and possibly the cuneate nucleus[45]) are the earliest stages where signals coming from multiple fiber types converge and integrate with each other. This allows for the possibility that mimicking realistic neural responses of small mixed-type afferent populations will result in naturalistic patterns of cortical activation[41], culminating in quasi-natural tactile percepts. However, despite the initial success of biomimetic approaches in hand amputees, which outperformed classical non-biomimetic stimulation patterns, this approach was never tested in lower-limb amputees. Moreover, it was evaluated while performing tasks of daily living or in more complex scenarios than a single user with a single-channel stimulation. Furthermore, we still do not understand how these patterns are transmitted and interpreted in the first layers of information processing along the somatosensory neuroaxis.

To investigate the neural signal transmission along the somatosensory axis, we develop a neuroprosthetic framework constituted by realistic in-silico modeling, pre-clinical animal validation, and clinical testing in human patients with implants (Fig. 1). Using this multifaceted approach, we are exploiting the architecture established by the development of validated model-based neurotechnology in human applications. Specifically, we first designed biomimetic neurostimulation strategies to restore somatosensory feedback using a realistic in-silico model of human afferent behavior (FootSim)[46]. This computational model can emulate the neural activity of the sensory afferents, innervating the plantar area of the human foot, in response to spatio-temporal skin deformations. It allowed us to design neurostimulation patterns that potentially mimic relevant temporal features of the natural touch coding during walking. In addition to developing stimulation paradigms, we addressed a significant challenge: understanding how specific artificial stimulation patterns are translated into neural signals and how these signals propagate along the somatosensory neuroaxis. With this goal, we stimulated the tibial nerves of decerebrated cats with cuff electrodes while simultaneously recording neural activity (in Dorsal Root Ganglion (DRG) with a 32-channel Utah array and in the spinal cord (L6) with a 32-channel shaft electrode). This setup allowed us to record and compare the electrically induced activity (in response to different patterns of nerve stimulation) with the response of neurons to mechanical touch. We validated this multifaceted approach through tests with three transfemoral amputees with implants in the tibial nerve. First, we compared the naturalness of the evoked artificial sensation using biomimetic and non-biomimetic encodings. Then, we implemented the biomimetic neurostimulation in a real-time, closed-loop neuroprosthetic leg, comparing its performance with respect to previously adopted neurostimulation strategies (linear and discrete neuromodulations). The patients' performance was assessed during ecological motor tasks (i.e., stairs walking task[47] and a motor-cognitive dual task[48]).

Both the animal and human experiments indicate that time-variant, biomimetic policies of artificial electrical stimulation should become the fundamental feature for the next generation of neuroprostheses, able to directly communicate physiologically plausible sensations to the brain.

## Results

To design an optimal neurostimulation strategy based on a bio-inspired computation, able to effectively convey somatosensation, we exploited a trifold framework including computational modeling, animal testing, and human clinical trial (Fig. 1). Computational modeling consisted of designing neurostimulation strategies to mimic the natural touch computation, shaped by a realistic in-silico model, called FootSim[46]. This model allows us to emulate the spatio-temporal dynamics of the natural touch code, considering all the tactile afferents innervating the plantar area of the foot. Experimental steps involved tests in animals where we showed that the biomimetic paradigm was transmitted through the somatosensory neuroaxis following the same neural dynamics from the periphery to the spinal cord. Neural responses evoked at multiple levels of the somatosensory neuroaxis showed higher similarity to the neural activity induced by natural touch compared to the standard stimulation methods, suggesting that these biomimetic patterns could potentially be the optimal encoding strategy for human neuroprosthetics. Indeed, we implemented the sensory encodings in a closed-loop neuroprosthetic leg, comparing both sensation naturalness and feedback performance in the context of everyday life activities.

### Biomimetic neurostimulation paradigms are designed by exploiting a realistic in-silico model of foot sole afferents (FootSim)

We used the computational model of foot sole cutaneous afferents (FootSim)[46] to design biomimetic stimulation strategies.

FootSim is able to emulate the spatio-temporal dynamics of the natural touch considering the activation of all tactile afferents innervating the plantar area of the foot[49]. This model is a plug-and-play tool,

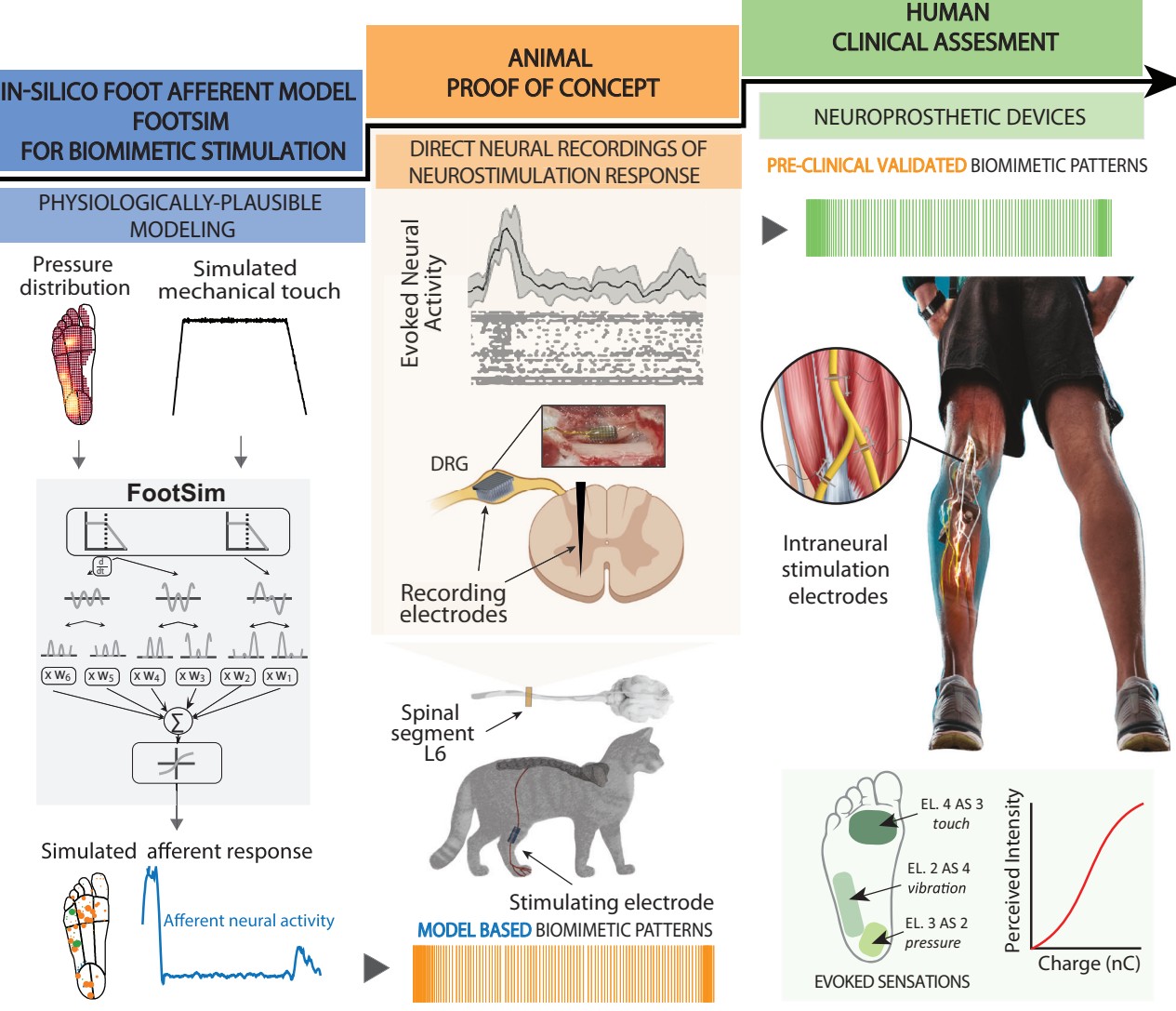

**Fig. 1 | Neuroscience-driven development of a biomimetic neuroprosthetic device.** The successful development of a somatosensory neuroprosthesis is based on three main pillars: (1) In-silico models of the biological sensory processing have to be exploited for emulating the natural neural activation of the nervous system to external tactile stimuli (blue segment); (2) animal proof of concept allows for experimental validation of the mechanisms behind the use of specific neurostimulation strategies defined with the use of modeling (orange segment); DRG–dorsal root ganglion. (3) A rigorous clinical validation of the biomimetic technology with implanted humans has to be performed in order to assess the functional outcomes in real-life scenarios (green segment). The results from the clinical trials will then allow us to collect relevant data exploitable for improving computational modeling. (Illustration credit: Giacomo Valle, ETH Zurich).

fitted on the human microneurography data, which models mechanical input from the external environment and gives the output corresponding neural afferent activity (Fig. 2a). While setting up the environment, the user is populating the foot sole arbitrarily, depending on the case and envisioned usage. The foot can be filled with a realistic or modified distribution of a specific type of afferent or, alternatively, with the complete population. Different mechanical stimuli could be applied. We have the capability to simulate either a single mechanical stimulus applied to a specific position on the plantar side of the foot or a scenario of a person walking. It can be achieved by extracting the pressure distribution across the entire foot sole at different time steps. (Fig. 2a left). The FootSim output can be structured in several forms. We can extract spike trains of a single afferent, of summed population activity, or spatially represent the activity of the afferents placed in the foot sole by coding their firing rates with the area of the circle (Fig. 2a right).

When designing the biomimetic patterns, we also followed the aim to unveil if the naturalness can be coded in the neural responses

specific to afferent types. We created five different scenarios by populating the foot sole with different types of afferents (Fig. 2b: FAI/FAII/SAI/SAII only) or with a complete population realistically existing in the human foot (Fig. 2b: FULL population). We applied a ramp-and-hold stimulus covering the whole foot sole by adding the environmental noise to mimic the imperfection of the realistic pressure stimuli (Fig. 2b black line). We calculated the peristimulus time histogram (PSTH), merging all afferent responses based on the scenario (Fig. 2b colored lines). We used smoothed PSTH values to modulate the stimulation frequency while keeping the amplitude constant and create biomimetic neurostimulation paradigms. (Fig. 2b: FAI/FAII/SAI/SAII/FULL biomimetic).

## The neurostimulation dynamics are transferred through somatosensory neuroaxis

We recorded intra-spinal neural response signals and activity in dorsal root ganglion (DRG) in two cats to be able to compare bio- and non-bio inspired stimulation patterns and study their transmission through

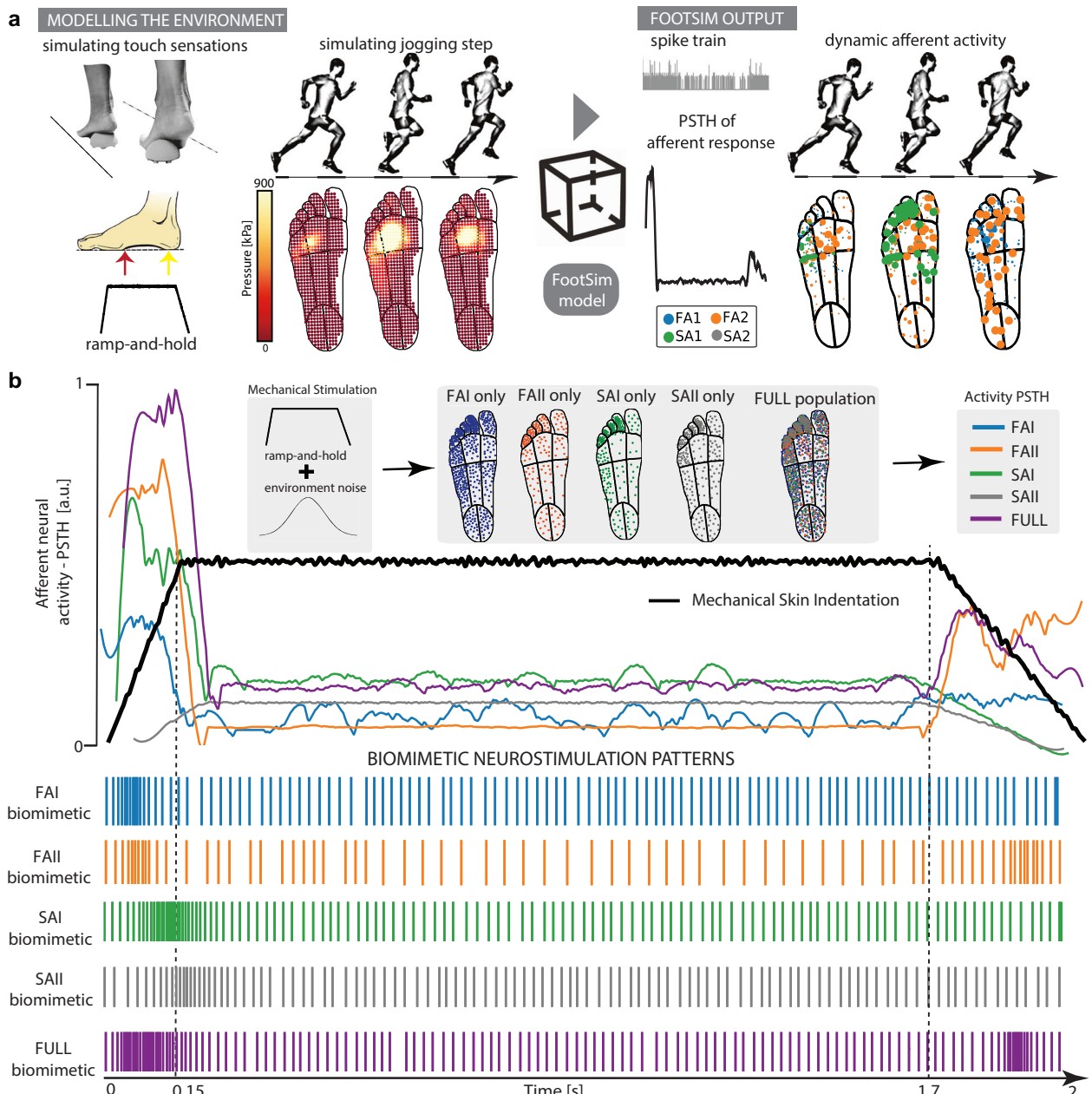

**Fig. 2 | Biomimetic neurostimulation patterns designed using a realistic in-silico model of foot sole afferents (FootSim). a** A schematic representation of practical use of the FootSim plug-and-play model. The environment is modeled as a single or continuous mechanical stimuli. The user can apply different types of stimulus or simulate the walking scenario, and the stimuli are given as input to the model in the form of pressure distribution across the foot sole. The output is an afferent neural response that can be presented in several ways: as a single spike train, spatially represented on the foot sole by matching afferent firing rate with the area of the circle placed on the position of afferent, or as a populational response with peristimulus time histogram (PSTH). **b** Foot sole is populated with a single type of afferent or with the whole realistic population (FAI/FAII/SAI/SAII population; FA-fast adapting; SA-slowly adapting; I/II−type 1 or type 2). We set the stimuli as a ramp-and-hold stimulus combined with the environmental noise and applied it on the whole foot area (black line). Neural responses of the whole applied population are given in the form of PSTH (colored lines). This was used as a function for the changes in frequency for defining biomimetic stimulating patterns. Amplitude remained constant in all biomimetic paradigms. All population distributions, afferent responses, and respective biomimetic stimulation patterns are color-coded: FAI: blue; FAII: orange; SAI: green; SAII: gray; FULL: purple.

somatosensory axes. Cats were decerebrated for enabling the analysis of only reflex responses, avoiding the signal interference with voluntary movements[50]. Also, this procedure allows the testing without the use of anesthesia, which could potentially alter the neural responses[51]. We implanted cuff electrodes on the tibial nerve for electrical stimulation and tuned the stimulation amplitude to be slightly above the threshold. Threshold was defined as an amplitude that clearly evoked potentials in the spinal cord in response to low-frequency stimulation. As multielectrode arrays appeared to be the best tool for investigating

the spinal cord processes[52], we extracted neural signals from a dorsoventral 32-channel linear probe implanted within the L6 spinal segment. Additionally, (Fig. 3a) we recorded neural signals in DRG at the L6 level with a UTAH array with 32 channels, as it contains the cell bodies and axons of sensory units from the periphery[53].

We tested the differences in neural dynamics that result from stimulating the tibial nerve with biomimetic paradigms and with a tonic 50 Hz pattern that is commonly used in neuroprosthetic applications. We performed multi-unit threshold crossing analysis to

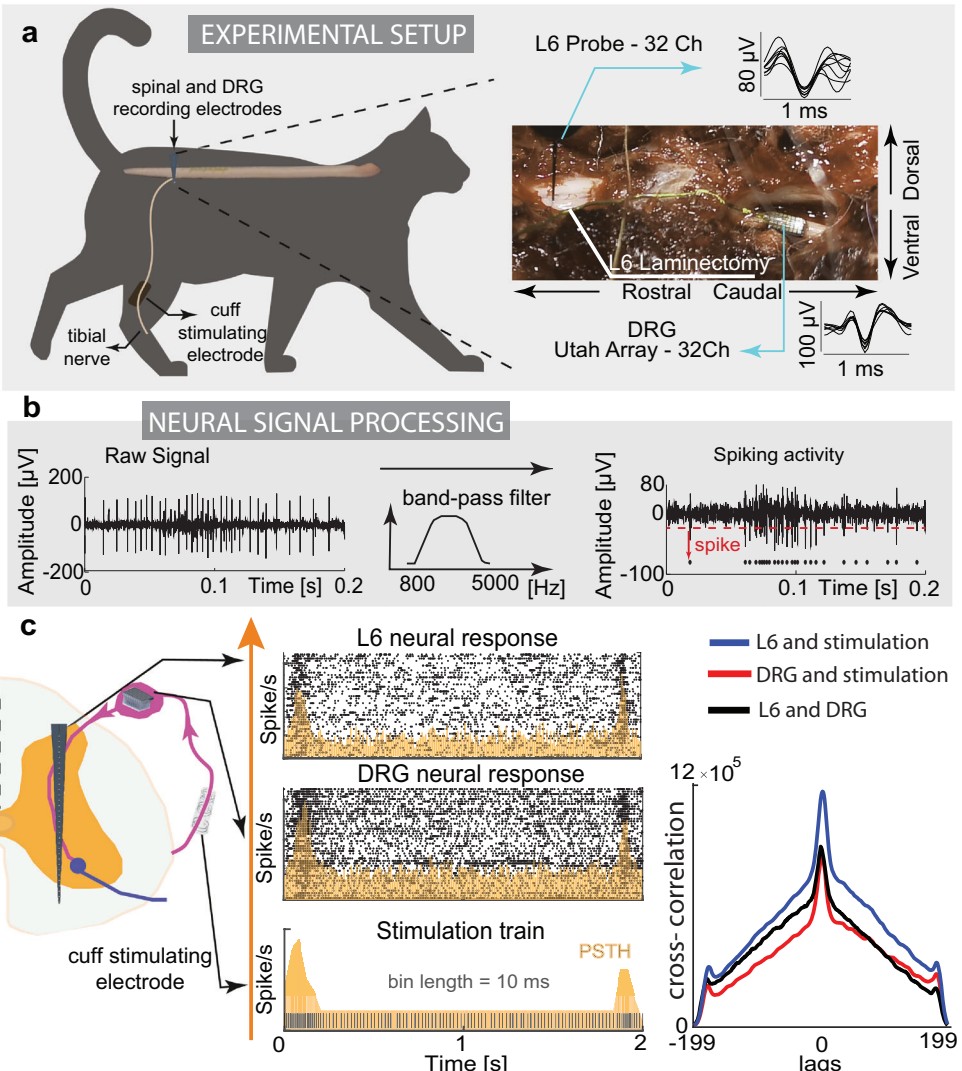

**Fig. 3 | Animal experiments for studying neural dynamics through neurostimulation. a** Decerebrated cat experimental setup. We stimulated the tibial nerve with a cuff electrode and recorded neural response on the spinal level; upper part: exposed L6 vertebrae and dorsal root ganglion (DRG) with examples of recorded neural spikes from spinal linear electrode probe and DRG UTAH array. **b** Obtaining multiunit neural activity. We filtered the signal to extract the spiking component and detect the neural action potentials using the thresholding algorithm. **c** left: Example of biomimetic stimulation paradigm and recorded response

signal in one channel of the spinal cord and DRG electrodes. Neural activity is presented and quantified with a raster plot (black dots) and peri-stimulus time histogram (PSTH, yellow, bin length 10 ms). Each row of the raster plot represents the response to a single biomimetic pattern (2 s), while each dot corresponds to an action potential. Right: We compared the PSTHs from different conditions and presented them using cross-correlation (blue: L6 neural response and stimulation pattern; red: DRG neural response and stimulation pattern; black: L6 and DRG neural responses). Source data are provided as a Source Data file.

identify the neural spiking activity (Fig. 3b), presented the results in the form of a raster plot, and quantified them using peri-stimulus time histogram (PSTH) (see Methods). The temporal dynamics of the neural activation pattern were highly correlated to the frequency of the neurostimulation train (Fig. 3c). We computed the cross-correlation of the PSTHs derived from the stimulation and neural responses recorded in the DRG and spinal cord. The cross-correlation values resulting when comparing stimulation shape, DRG signal and spinal neural response are high. It confirms the hypothesis that the biomimetic pattern of activation was transported to the DRG and spinal cord, maintaining the same spatiotemporal neural dynamics. In other words, by looking at the PSTH of single electrode channels, we observed that multiple peripheral afferent responses followed the biomimetic pattern and thus encoded the artificial tactile information.

This evidence strongly supports the notion that electrical neural stimulation can serve as a highly efficient tool for generating artificial patterns of neural activations that can be effectively communicated to

the upper regions of the somatosensory system. Furthermore, biomimetic patterns of neurostimulation, induced at the peripheral nerve level, evoked very similar spatiotemporal neural dynamics in the spinal cord.

### The neural response evoked by biomimetic stimulation is more similar to the mechanically induced activity than the one produced by tonic electrical stimulation

We base our hypotheses of evoking close-to-natural perception with biomimetic stimulation on the ability to code and replicate natural neural patterns. We recorded and compared the neural responses in the DRG and spinal cord resulting from different types of electrical stimulation with the naturally induced neural activity produced by touching the cat's leg with the cotton bud.

Comparing the characteristics of the electrically-evoked neural dynamics resulting from applied biomimetic, non-biomimetic, and natural stimulation confirmed previous theories[31,37,39]. We confirmed

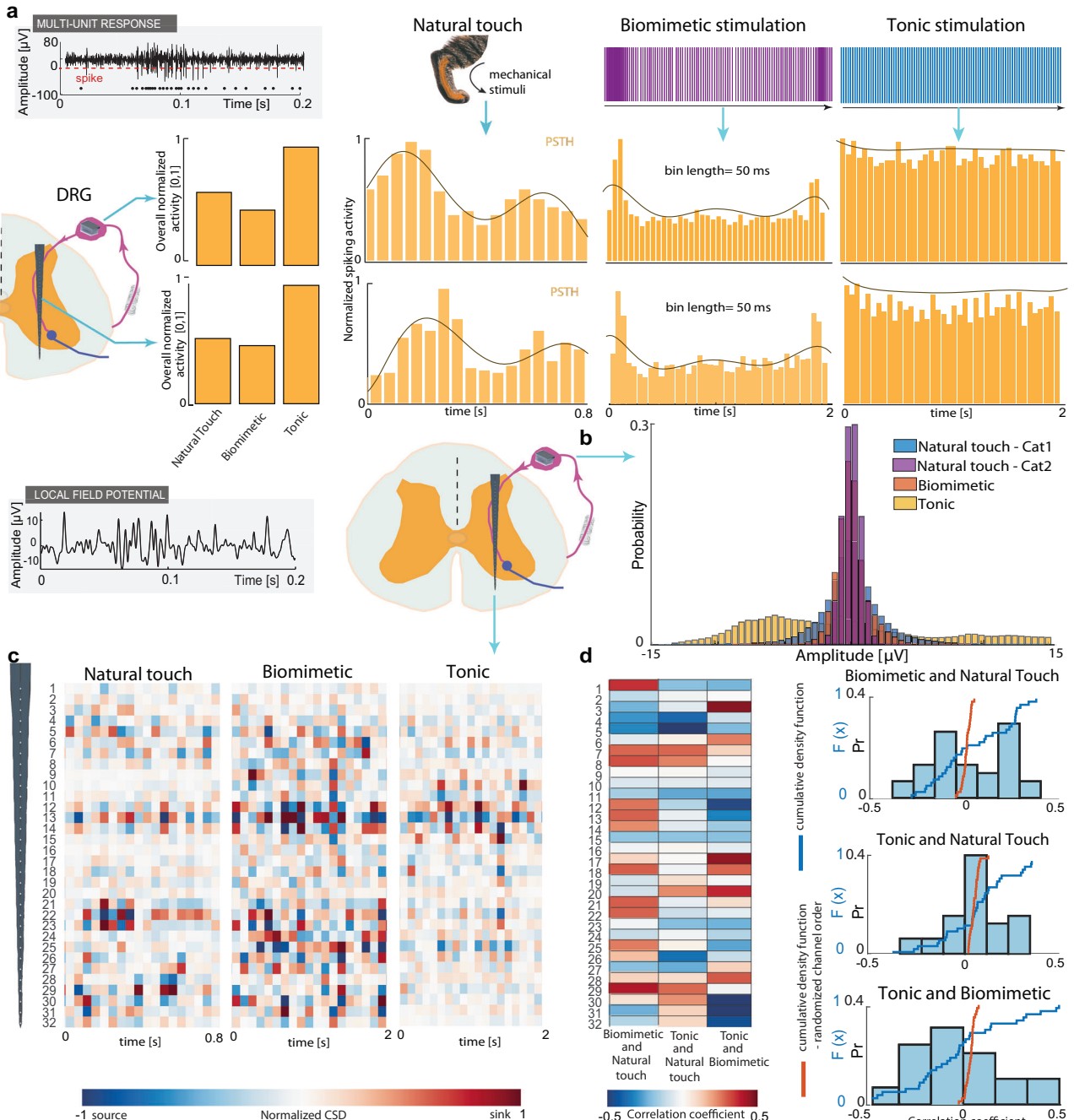

**Fig. 4 | Neural response with biomimetic stimulation is more similar to the response to natural touch than to tonic stimulation. a** Comparing multiunit neural activity as a response to natural touch, biomimetic, or tonic (50 Hz) stimulation. We compared the signal recorded in DRG (dorsal root ganglion) and spinal cord (level L6). The overall amount of neural activity during each condition is summed, normalized, and presented with bars for comparison (left). Examples of spiking activity over time during each condition are presented using a peri-stimulus time histogram (PSTH) (right) with a time bin of 50 ms. Brown lines represent the envelope of neural activity. **b** Comparing local field potential (LFP) recorded in DRG resulting from natural touch in biomimetic and tonic stimulation. Another natural touch response, recorded in cat 2, was added to the analysis. We compared the distribution of LFP amplitude values using the Kullback–Leibler divergence metric. **c** Comparing current source density (CSD) calculated from LFP recorded in the spinal cord resulting from natural touch, biomimetic, or tonic stimulation. CSD is normalized for each condition and presented along the length of the electrode with 100 ms bin for electrical stimulation and 40 ms for natural touch condition. **d** Left: correlation of CSD between biomimetic/tonic and natural touch condition and biomimetic and tonic stimulation, channel by channel, color-coded. Right: Histogram and cumulative distribution function (cdf) of the correlation coefficient values resulting from comparing biomimetic/tonic stimulation and natural touch condition (top/middle) and biomimetic and tonic stimulation (bottom). The blue line represents the cdf when the recording channels are matched and compared. The red line corresponds to cdf when channels are randomly shuffled and compared. Source data are provided as a Source Data file.

that the temporal pattern of the evoked-response exploiting biomimetic neurostimulation encoding was more similar to the one generated by mechanical stimulation of the skin of the animal than the one induced with tonic stimulation. We represented multi-unit spiking activity with PSTH (Fig. 4a). We calculated mean neural activity produced during the period of electrical or natural stimuli for estimating the overall amount of information occupying the spinal cord and DRG. We normalized the PSTH activity of each condition to be in the range [0,1] by dividing each point of the signal with the maximal signal value (Fig. 4a, right). We summed the activity for each condition

and divided it by the maximum activity between the three conditions. The bars show the overall normalized activity for the three conditions (Fig. 4a, right). The metric is inversely related to the variance of PSTH values across the condition. Natural touch and biomimetic stimulation resulted in similar values, while tonic stimulation inducted much higher activity in the spinal cord and DRG. We hypothesize that during tonic stimulation, the spinal neural networks are overwhelmed with constantly induced synchronized information, which can cause the paresthesia that is often perceived with commonly used neuromodulation paradigms[35,54]. Analogously, the presented similarity of neural natural activity and the one induced with biomimetic stimulation could explain why biomimetic stimulation is perceived as more natural. The shape of the PSTH and its envelope gave an insight into how neural activity is changing during the period of stimulation (natural, tonic, or biomimetic). Biomimetic stimulation produces more similar activity than the natural touch compared to tonic stimulation (Fig. 4a, right). The encoded message is represented in the neural dynamics of activation. Results reveal that the information produced with biomimetic stimulation matches better the natural touch neural coding then the commonly used tonic stimulation paradigm.

Local field potential (LFP) reflects the summed activity of a small population of neurons represented by their extracellular potentials[55] and they capture network dynamics[56]. We performed the analysis of the trigger-averaged LFP signal for different stimulating conditions. We extracted the DRG most active channels where clear LFPs were visible and investigated their amplitude variations. More in detail, we compared the amplitude distribution of recorded LFP using the Kullblack–Leibler divergence (KL) metric (Fig. 4b, see Methods). It reflects how one probability distribution diverges from the second, expected probability distribution. The larger the KL divergence, the more dissimilar the two distributions are. In our case, the expected distribution is the one arising from the natural touch. Therefore, we compared biomimetic and tonic stimulation responses to the one caused by natural touch (biomimetic and natural KL = 0.26; tonic and natural KL = 2.06). An addition, we tested natural touch conditions in one more cat to investigate the cross-subject similarities of neural dynamics (KL = 0.66). This evidence suggests that the naturally evoked response follows a specific, potentially generalizable trend rather than being completely individual.

Current source density (CSD) is a technique for analyzing the extracellular current flow generated by the activity of neurons within a population of neurons. It can estimate the location and magnitude of current sources and sinks that contribute to the measured electrical signals. Therefore, we used it for comparing the spatial distribution of neural activity within a population of neurons along the array in the gray matter of the spinal cord. We present the CSD estimated using local field potentials induced with biomimetic, tonic electrical stimulation, or natural touch (Fig. 4c). By visually inspecting the spatial distribution of sinks and sources along the spinal axes and comparing the overall Pearson correlation coefficients between CSDs resulting from different conditions, we can conclude that naturally induced touch response was more similar to the neural signal resulting from biomimetic stimulation (correlation coefficient 0.11, $p = 0.005$, $\alpha = 0.05$, confidence interval = [0.03, 0.19]) than to the one produced with constant, 50 Hz electrical stimulation (correlation coefficient = $-0.03$, $p = 0.344$, $\alpha = 0.05$, confidence interval = [$-0.11$, 0.04]). Biomimetic and tonic stimulation showed negative CSD similarity (correlation coefficient = $-0.13$, $p = 0.001$, $\alpha = 0.05$, confidence interval = [$-0.2$, $-0.05$]). We presented a color-coded channel-by-channel comparison (Pearson correlation coefficient) of the resulting CSDs along the spinal electrode (Fig. 4d, left) and quantified the results with a histogram and resulting cumulative distribution function (CDF) (Fig. 4d, right). The CDF describes the probability that a random variable takes on a value less than or equal to a specified number and was used to compare distributions reflecting the comparison between CSD in different

conditions. Tonic stimulation and natural touch produce neural responses with a correlation coefficient very close to 0 in most of the channels, while that coefficient is higher for comparison between natural touch and biomimetic stimulation. In order to verify that this similarity is not produced by chance, we randomized the order of the channels in biomimetic and tonic electrical stimulation conditions and compared the recordings with the response of natural touch. It produced a correlation close to 0 for every electrode channel, confirming the validity of the used analyses.

Furthermore, we analyzed how much the neural signal is changing along the transversal spinal axes. We compared the Pearson correlation coefficients between the LFP in the first channel of the intraspinal array and all the other channels (Fig. S1). In the natural touch condition, the similarity between the neural activity is high in the first few channels (2nd channel correlation coefficient is 0.37, 3rd 0.34, 4th 0.1; $p < 0.001$, $\alpha = 0.05$), and it is diminished when looking at more ventral recordings (less than 0.1 correlation coefficient, leading to median correlation coefficient value of 0.04, 25th percentile 0.02 and 75th percentile 0.05). When a nerve was electrically stimulated, the similarity between the neural activity recorded with the different channels through a spinal array is high (FA1: median 0.94, 25th perc. 0.90, 75th perc. 0.95; FA2: median 0.86, 25th perc. 0.80, 75th perc. 0.92; SA1: median 0.87, 25th perc. 0.83, 75th perc. 0.89; SA2: median 0.92, 25th perc. 0.88, 75th perc. 0.93; Fig. S1). The biomimetic neurostimulation elicited less similarity along the spinal axes than tonic stimulation (biomimetic FULL: median 0.6, 25th perc. 0.45, 75th perc. 0.67; 50 Hz tonic: median 0.88, 25th perc. 0.83, 75th perc. 0.9; Fig. S1). Full population biomimetic pattern showed to be the more promising one compared to the paradigms created by mimicking response of specific afferent types. Despite being significantly different from the natural touch, biomimetic stimulation based on aggregate population of afferent responses shares a striking similarity with it, setting it significantly apart from the tonic, 50 Hz stimulation ($p$ values: FA1- FULL biom: <0.001; FA2- FULL biom: 0.001; SA1- FULL biom: 0.01; SA2- FULL biom: <0.001; 50 Hz-FULL biom: 0.001; Natural- FULL biom: 0.035; 50 Hz-Natural: $p < 0.001$; significance level 1%, $\chi^2 = 187.4$).

## Biomimetic neurostimulation evokes more natural sensations than non-biomimetic neurostimulation paradigms

To test the functional implication of using biomimetic neurostimulations, we implemented and tested them in a human clinical trial. Our first aim was to validate the biomimetic neurostimulation encoding by assessing the quality of the evoked sensations. Then, a real-time neurorobotic device exploring biomimetic encoding strategies has to be compared to devices with previously adopted encoding approaches in terms of functional performances. We implanted three transfemoral amputees (Table S1) with TIME electrodes in the tibial branch of the sciatic nerve (Fig. 5a). After conducting a sensation characterization procedure, where all the 56 electrode active sites were tested, a subgroup of electrode channels were selected for this evaluation. Specifically, groups of active sites eliciting sensations located in the frontal, central, lateral metatarsus, and heel were identified (Fig. 5b and Fig. S2). In this way, the selected channels were electrically activating different groups of mixed afferents with projecting fields in different areas of the phantom foot (with different distributions of innervating fibers).

Then, multiple strategies, encoding a mechanical skin indentation, were adopted to deliver neurostimulation trains through each selected channel of the intraneural implants (Fig. 5c). The participants were asked to report the perceived sensation naturalness using a visual analog scale (VAS) between 0 (totally non-natural sensation) and 5 (totally natural sensation–skin indentation)[39,57]. In all three implanted participants and across all the active sites tested (with different projected fields), the biomimetic neurostimulation patterns elicited sensations more natural than the linear neurostimulation encoding

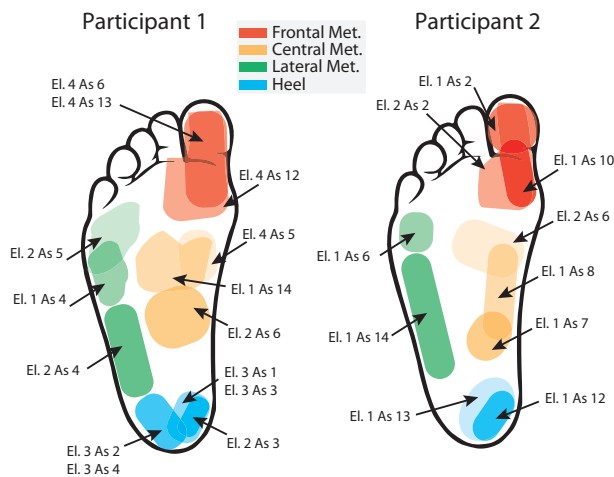

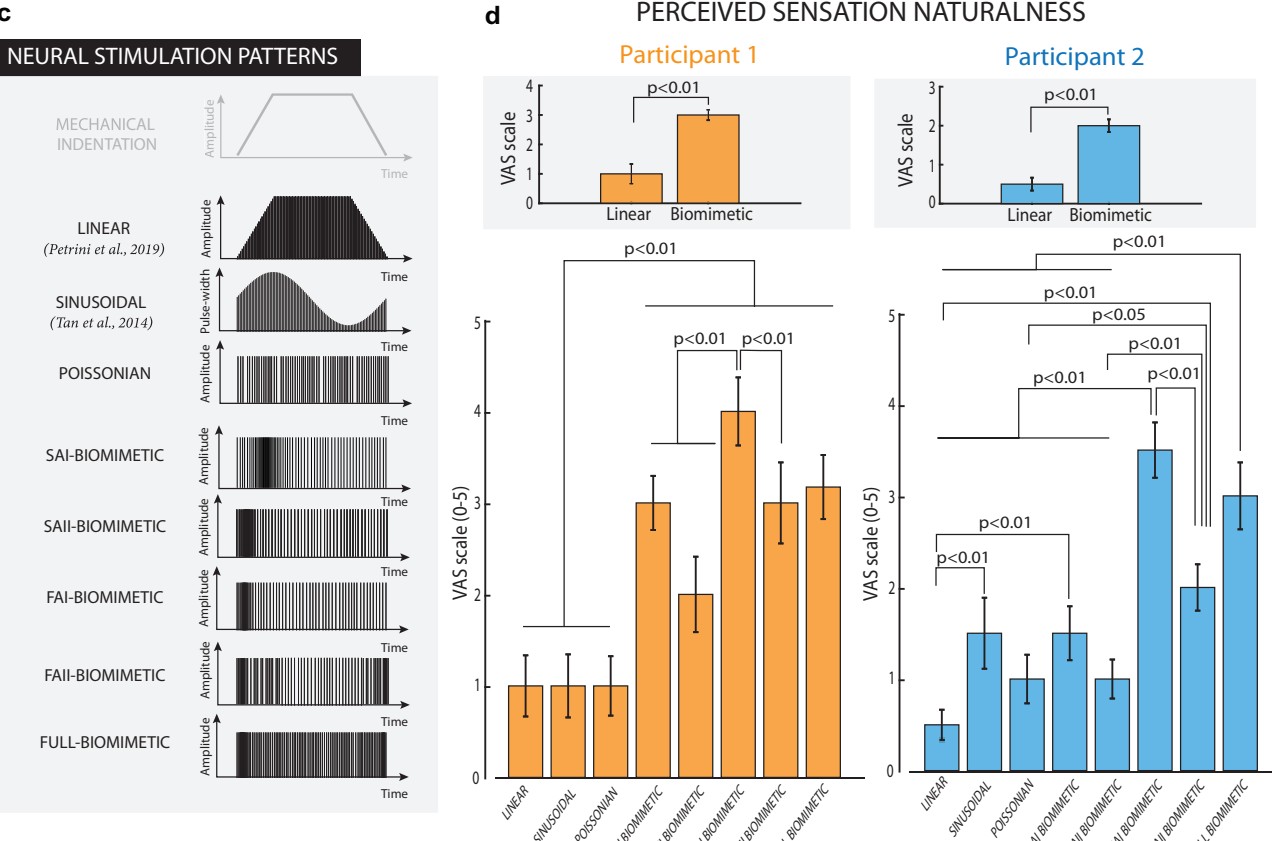

**Fig. 5 | Biomimetic neurostimulations evoke more natural perceptions in implanted humans than non-biomimetic approaches. a** Individuals with lower-limb amputation were implanted with TIME in their tibial nerves. The multichannel electrodes were used to directly stimulate the peripheral nerves evoking sensation directly onto the phantom foot. Segments of the panel upper part taken from with permission from[106]. **b** Projective fields map of two implanted participants (1 and 2) related to the active sites (AS) adopted to electrically stimulate the nerves. Different colors show the four main regions of the phantom foot (Frontal, Lateral Central Metatarsus, and Heel). **c** Biomimetic and non-biomimetic neurostimulation strategies adopted for encoding a mechanical indentation of the foot sole. Linear

neurostimulation is taken from and Sinusoidal neurostimulation by[10].
**d** Naturalness ratings (VAS scale, displayed on a scale of 0–5) of the perceived sensation elicited by exploiting different stimulation strategies in two participants. Insets: Group comparison between linear vs biomimetic stimulations We compared the conditions using the Kruskal–Wallis test ($n = 75$ stimulation repetitions per condition for Participant 1 and $n = 33$ stimulation repetitions per condition for Participant 2). A post-hoc correction was executed. BIOM-LIN: S1: $p < 0.001$, $f = 0.62$; S2: $p < 0.001$, $f = 0.89$) Data are presented as median values ± standard deviation. Source data are provided as a Source Data file.

$(3 \pm 0.18$ with Biomimetic compared to $1 \pm 0.35$ in Linear for S1, $2 \pm 0.16$ with Biomimetic compared to $0.5 \pm 0.17$ in linear for S2, and $2 \pm 0.36$ with Biomimetic compared to $1 \pm 0.18$ in linear for S3 across all electrode tested) (S1: $p < 0.001$, $f = 0.62$; S2: $p < 0.001$, $f = 0.89$; S3: $p = 0.0026$, $f = 1.01$) (Fig. 5d and Fig. S3) that was previously adopted in multiple neuroprosthetic applications[8,26]. Moreover, biomimetic-based encodings often resulted in more natural perceived sensations compared to both sinusoidal (pulse width-variant) (S1: $p < 0.001$ for SAI, FAI, FAII, FULL Biomimetic, $p = 0.002$ for SAII; S2: $p = 0.011$ for FAI, $p = 0.016$ for FULL Biomimetic; S3: $p = 0.035$ for FAII, $p = 0.019$ for FULL Biomimetic) and Poisson (frequency-variant) neurostimulation strategies (S1: $p < 0.001$ for SAI, SAII, FAI, FAII, FULL Biomimetic; S2: $p < 0.001$ for FAI, FULL Biomimetic, $p = 0.039$ for FAII; S3: $p = 0.008$ for SAII, $p = 0.001$ for FAI, FULL Biomimetic), indicating the importance of inducing a neural activation dynamic mimicking the natural biological code.

Notably, although multiple biomimetic-like paradigms have been tested (SAI-, SAII-, FAI-, FAII-like, and Full biomimetic), none of them proved to be better. Although biomimetic stimulation always elicited more natural sensations than one parameter adopted encoding, analyzing the results per location in both participants (Fig. S3) did not show any clear evidence of an optimal biomimetic encoding schema. This was probably caused by the different composition of the fibers activated by the electrode channels in the different foot regions[58]. In fact, the perceived areas were different according to the active site selected to stimulate, indicating a different group of mixed afferents recruited by the neurostimulation. We hypothesized that not only the proportion of SA and FA fibers is relevant but also their role in encoding touch information in that specific region.

These findings highlighted how biomimicry is a fundamental feature of electrical neural stimulation for successfully restoring more natural somatosensory information.

## Biomimetic neurostimulation on a neuro-robotic device allows for higher mobility and a reduced mental workload

Aiming to develop a neuroprosthetic device able to replace the sensory-motor functions of a natural limb as much as possible, this biomimetic neurostimulation was then implemented in a real-time robotic system. This wearable system was composed of (i) a sensorized insole with multiple pressure sensors, (ii) a microprocessor-based prosthetic knee with a compliant foot (Ossur, Iceland), (iii) a portable microcontroller programmed with biomimetic sensory encoding algorithms, (iv) a multichannel neurostimulator; (v) intraneural electrodes implanted in the peripheral nerves (TIMEs).

The neuroprosthetic device recorded pressure information in real-time from the wearable sensors while the patient was walking and converting them into patterns of biomimetic neurostimulation delivered through the TIMEs (see Method for implementation details). In this way, the users were able to perceive natural somatotopic sensations directly from the prosthetic leg without any perceived delay.

After the implementation, we assessed the effects of exploiting the biomimetic encoding (BIOM) in a neuro-robotic device compared to a linear (LIM) or a time-discrete (DISC) neurostimulation strategy. In the LIN, the sensors' readouts were converted in neurostimulation trains following a linear relationship between applied pressure and injected charge[26]. In the case of DISC, short-lasting, low-intensity electrical stimulation trains were delivered synchronously with gait-phase transitions[59,60]. Also, the condition without the use of any neural feedback (NF) was included in the motor paradigms as a control condition.

The neuroprosthetic users were thus asked to perform two ecological motor tasks: Stairs Task (ST)[47] and Cognitive Double Task (CDT)[48].

In ST, results indicated that, when exploiting biomimetic neurostimulation in a neuro-robotic leg, both users improved their walking speed $(4.9 \pm 0.1$ for S1 and $4.3 \pm 0.4$ for S2 laps/session) compared to LIN $(4.5 \pm 0.1$, $p < 0.001$ for S1 and $3.8 \pm 0.1$, $p = 0.047$ for S2 laps/session), DISC $(4.6 \pm 0.1$, $p = 0.004$ for S1 and $3.6 \pm 0.1$, $p = 0.047$ for S2 laps/session) and NF $(4.3 \pm 0.1$, $p < 0.001$ for S1 and $3.5 \pm 0.1$, $p < 0.001$ for S2 laps/session) conditions (total effect size, $f = 2.14$ for S1, $f = 1.5$ for S2) (Fig. 6a). Interestingly, also the self-reported confidence (VAS scale 0–10) in walking on stairs was increased, when the participants were exploiting the neuroprosthetic device with biomimetic neurofeedback $(9.75 \pm 0.26$ for S1 and $6 \pm 0.3$ for S2) compared to LIN $(8.75 \pm 0.62$, $p = 0.015$ for S1 and $5.37 \pm 0.23$, $p = 0.014$ for S2), DISC $(7.83 \pm 0.39$, $p < 0.001$ for S1 and $5.17 \pm 0.25$, $p < 0.001$ for S2) and NF $(6.67 \pm 0.49$, $p < 0.001$ for S1 and $3.83 \pm 0.25$, $p < 0.001$ for S2) conditions (total effect size, $f = 2.57$ for S1, $f = 3.09$ for S2) (Fig. 6a).

In the CDT, both participants showed a higher mental accuracy in BIOM compared to the other conditions ($p = 0.004$ for NF, $p = 0.016$ for LIN, $p = 0.028$ for DISC in S1; $p = 0.044$ for LIN, $p < 0.001$ for DISC and NF in S2; total effect size, $f = 0.57$ for S1 and $f = 0.97$ for S2), while maintaining the same walking speed. In particular, the mental accuracy of S1 was $76 \pm 16\%$ in BIOM, $58 \pm 20\%$ in LIN, $58 \pm 11\%$ in DISC, and $52 \pm 17\%$ in NF, while in S2 $94 \pm 9.6\%$ in BIOM, $72 \pm 17\%$ in LIN, $50 \pm 37\%$ in DISC and $48 \pm 14\%$ in NF (Fig. 6b). Notably, the walking speed was always higher in the feedback conditions compared to NF for S2 ($p < 0.001$ for DISC, LIN, and BIOM) and in BIOM ($p < 0.001$) and in LIN ($p = 0.002$) for S1 (total effect size, $f = 0.91$ for S1 and $f = 1.13$ for S2). As expected, without adding a secondary task, no difference was observed in the walking speed among the conditions in both participants ($p > 0.08$, $f = 0.50$, for S1 and $p = 0.37$, $f = 0.36$ for S2; Fig. S4). Analyzing the spelling accuracy, both participants showed low Spearman's rank correlation coefficients (between 0.03 and 0.5) across sessions, suggesting no learning or initial habituation effect on their performance (Fig. S5).

These findings indicated a higher decrease in mental workload while the users were performing two tasks simultaneously (one motor and one cognitive) at the moment that a more bio-inspired neural stimulation was exploited in a neuro-robotic device.

## Discussion

### Multi-level approach for designing stimulation strategies that would minimize paresthesia sensations

In this study, we designed, developed, and tested a neuro-robotic device exploiting model-based biomimetic neurostimulations in people with limb amputation. Due to a multilevel framework, it was possible to design and test effective bio-inspired neurostimulation paradigms to elicit more natural feelings and better understand the reasoning behind the use of biomimetic approaches in the neuroprosthetic field. Indeed, thanks to realistic in-silico modeling of the foot touch coding, precise neural stimulation patterns were defined that accurately emulate the firing of the cutaneous mechanoreceptors. Single-fiber (SAI, SAII, FAI, FAII) and mixed-fiber (FULL) type patterns have been implemented to encode a mechanical skin indentation via our neural stimulation policy. We modulated the stimulation frequency based on the fiber dynamics of activation since it showed to be beneficial for shaping the artificial touch for bionic limbs[61].

### Comparing neural responses induced with natural touch and electrical nerve stimulation

The designed animal experiments allowed us to compare the neural dynamics as a response to natural touch, biomimetic, or tonic electrical stimulation. In this work, we used cats as an animal model due to their similar peripheral sensory fiber activation to humans during locomotion[62]. Decerebrated cats' experiments allowed us to objectively inspect the propagation of biomimetic paradigms from the periphery, which was now known a priori, in a controlled and undisturbed manner. This enabled the unique experimental setup for the comparison of the features of neural response to the natural touch and

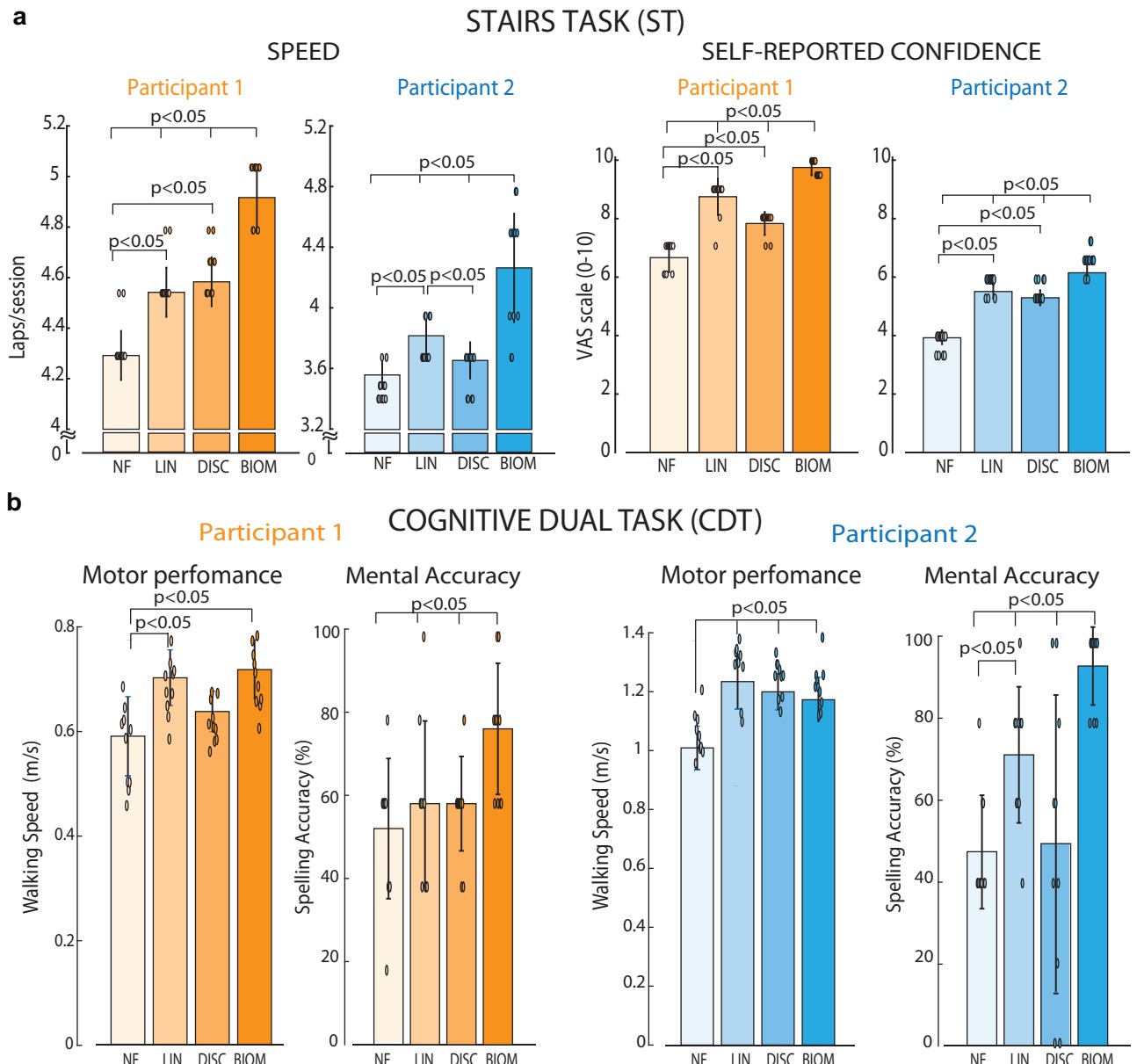

**Fig. 6 | Real-time biomimetic neural feedback allows for higher speed and lower cognitive workload while walking. a** Speed (Laps/session) and self-reported confidence (VAS Scale 0–10) were measured in ST ($n = 12$ task repetitions for both participants). **b** Motor performance (Walking Speed–m/s) and Mental Accuracy (Spelling Accuracy–%) of Participants 1 and 2 in the Cognitive Dual Task (CDT) ($n = 10$ task repetitions for both participants). In both tasks, conditions are NF (No Feedback), LIN (Linear Neurostimulation), DISC (Discrete Neurostimulation), and BIOM (Biomimetic Neurostimulation). The ellipses overlapped with the bar plot represent single data points. Data are presented as mean values ± standard deviation. We compared the conditions using the Kruskal–Wallis test. A post-hoc correction was executed. Walking speed: BIOM-LIN: S1: $p < 0.001$, S2: $p = 0.047$; BIOM-DISC: S1: $p = 0.004$, S2: $p = 0.047$; BIOM-NF: S1,S2: $p < 0.001$ (total effect size, $f = 2.14$ for S1, $f = 1.5$ for S2) Confidence: BIOM-LIN: S1: $p = 0.015$, S2: $p = 0.014$; BIOM-DISC: S1, S2: $p < 0.001$; BIOM-NF: S1,S2: $p < 0.001$ (total effect size, $f = 2.57$ for S1, $f = 3.09$ for S2). Source data are provided as a Source Data file.

different stimulation paradigms at the spinal level without interfering with descendent signals that would be difficult to disentangle. Using the decerebration method, we also excluded the use of anesthesia as it can alter the neural signals and eliminate interference with movements. In these very complex, 12–15 h long, animal experiments, our main goal was to understand if the more sophisticated stimulation elicits a cumulative neural signal more similar to natural ones than the "classic" tonic stimulation at the spinal level. In these pioneering experiments, we were touching latero-caudal leg area involving the heel, which is innervated by a tibial and common sciatic nerve. These are the parts of the same common nerves going to the paw sole also. By doing so, we activated the afferent types that innervate the human foot

sole, as defined biomimetic patterns are developed on the basis of their activity. Our goal was to make a proof of concept of recording the "gross" signal elicited by touch and electrostimulation and then analyze its cumulative features at the population level. The recordings in decerebrated cats via multiple neural interfaces along their somatosensory neuroaxis (somatic nerve, DRG, and spinal cord) showed that biomimetic neurostimulations evoked spatio-temporal characteristics of the afferents' response more similar to those naturally induced one than tonic stimulation. These biomimetic patterns have significantly smaller synchronized activity in spinal circuits compared to the tonic stimulation. This highly increased activity can saturate the circuits and limit the possibility of perceiving touch sensations restored with

electrical stimulation[63]. This is clear evidence of the effect of bio-inspired stimulation dynamics on neural afferent activation, showing the possibility of artificially encoding natural sensory messages into the nervous system. Indeed, previous research has hypothesized the adoption of complex spatiotemporal patterns mimicking natural peripheral afferent activity[31,37]. This approach was also proposed for cortical activity modulation using intracortical microstimulation (ICMS) to convey feedback of touch[40,64] or of the entire movement trajectory (natural proprioceptive sensation)[65]. Likewise, they also assumed that exploiting an ICMS interface that mimics natural sensations would be faster and ultimately more effective than learning arbitrary associations with unnatural sensations or arbitrarily modulated ICMS[66]. Our study validates these hypotheses on the use of biomimetic encoding in PNS neuroprostheses. However, our experimental setup was focused on understanding the first layer of processing information coming from the periphery to the spinal cord. Going higher in this direction, cortical responses (LFPs) to biomimetic peripheral nerve stimulation using interfascicular electrodes have been recently measured in a monkey[67]. The authors showed that constant frequency stimulation produced continual phase locking, whereas biomimetic stimulation produced gamma enhancement throughout the stimulus, phase locked only at the onset and release of the stimulus. This cortical response has been described as an "Appropriate Response in the gamma band" (ARγ). Regarding the sensory restoration in bionics, multichannel biomimetic ICMS provided high-resolution force feedback[40] and more localizable sensations[68] in implanted humans. We believe future experimental work should extend these findings by investigating neural processes caused by electrical stimulation in the gracilis nucleus (or cuneate for the upper limb), thalamus, or the somatosensory cortex, particularly in humans.

## Biomimetic stimulation in neuromodulation devices is beneficial both for the perceived sensation and its functionality

These biomimetic neurostimulation strategies were tested in three human participants with transfemoral amputation implanted in their leg nerves with intraneural electrodes. All the participants reported feeling more natural sensations, when stimulated with biomimetic encodings with respect to standard neuromodulation patterns from every stimulation channel on the electrodes. Neural stimulation gradually recruits all the sensory afferents within a fascicle[69,70] depending on both distance from the electrode (threshold proportional to the square of distance) and afferents' diameter (threshold proportional to 1/square root of fiber diameter)[71]. Therefore, each stimulation pulse delivered through the active site is likely to recruit a mix of sensory afferent types, even if clustered[72]. For this reason, how many and what tactile afferents will be stimulated by a given stimulation pattern through a specific electrode is unknown a priori. This might be the reason why different types of biomimetic encoding were reported as more natural by the participants according to the perceived foot location (Fig. S3) and, therefore, to the clusters of recruited afferents. This phenomenon can also reveal the typology of sensation reported, while specific types of afferents were activated by neurostimulation[36,73] (flutter, vibration, touch). Similarly, it could explain why with simpler encoding (at the threshold level), the electrically evoked sensation can be sometimes reported as natural[8,10,74].

Finally, we implemented these algorithms in robotic prosthetic devices in real-time, comparing their functional performance with previously proposed technologies. Biomimetic neuroprosthetic legs allowed for faster stair walking and a decreased mental workload in a double-task paradigm in both participants. These findings demonstrated that biomimetic encoding is relevant for device functionality and thus enhances the beneficial effect of this intervention. In particular, a significant boost in mobility on a difficult everyday life task such as walking upstairs is very relevant for people with lower-limb amputation. This improvement is likely connected to reported higher confidence in the prosthetic leg with biomimetic sensory feedback[47]. The amputee is able to sense the position of their prosthesis with respect to the ground more effectively, which results in a faster transition from heel strike during walking[75]. Confidence and mobility have been previously proposed as the clearest and simplest measures of the impact of sensory feedback on gait[47]. Commercial microprocessor-controlled knees improve participants' self-selected walking speed by about 8% compared to mechanically passive devices[76]. In this study, we showed that the speed of participants in stair tasks while using a microprocessor-controlled knee (RHEO KNEE XC) was improved even more by biomimetic sensory feedback (>13% in S1 and >22% in S2). Regarding the self-confidence results, many studies have reported decreased self-confidence in walking for lower-limb amputees and its strong relationship with mobility and walking abilities[77-79]. Moreover, previous studies with upper-limb amputees have then shown that sensory feedback increased participants' confidence (i.e., self-efficacy) and was directly correlated with prosthesis accuracy in functional tasks[80,81].

The CDT represented a real-life scenario of multiple simultaneous tasks. It allowed us to obtain an objective measure of the better cognitive integration of the prosthesis with biomimetic neurostimulation[22,48], with both amputees improving their mental accuracy. In addition to our results, previous studies have shown preliminary improvements in manual dexterity and object recognition in upper-limb amputees via robotic hand prostheses[38,39]. Nevertheless, there are noticeable differences to consider between upper and lower-limb prosthetics in the design of synthetic sensory feedback. First, the sciatic nerve, which innervates the foot and lower leg, is more than twice as large as the median and ulnar nerves, which innervate the fingers and palms, and it is also difficult to reach through the big leg muscles during the surgery. The density and placement of the receptors are different between the upper and lower extremities (considered in TouchSim and FootSim). Upper-limb amputees can use their intact hands for almost all activities, while leg amputees cannot ambulate without a prosthesis. A failed manipulation can lead to a broken glass, whereas a failed step could lead to a dangerous fall. Finally, more proximal levels of amputation require more complexity in the sensory feedback restoration, as larger amounts of information need to be transmitted. These feedback specifications need to be specifically considered for the different amputation types.

## Limitations of the study

During the animal experiments, ideally, we would apply the mechanical stimuli replicating natural touch on the cat's paw in a controlled way, using a robotic setup that applies a constant, predefined pressure to elicit robust neural response along the somatosensory neuroaxis. However, that procedure is expensive, complex, and above all very time-consuming, therefore was impracticable in our proof-of-concept experiments. Future tests should examine the comparison of PNS with different types of natural tactile stimulation.

The FootSim model used for creating biomimetic paradigms did not incorporate shear forces or lateral sliding but simulated them as quasi-continuous stress. This simplification implies reduced accuracy of predicting the SA2 type afferents' responses, which transmit the information about skin stretch.

Even though we tested the biomimetic patterns in the closed-loop neuroprosthetic system, these paradigms were based on the model outputs to run offline, while we believe that the stimulation strategies should be defined in real-time from the output of the model. Engineering efforts are needed to make this neuroprosthetic system fully functional in real-time and, in the future, also fully implantable. To complete the assessment of this biomimetic neuroprosthesis, future clinical trials should include quantitative analyses of gait pattern changes[47,82] fatigue and neurophysiological measures, which are critical aspects for the long-term acceptance of these devices. Moreover,

while the assessment of naturalness using a single-item VAS is informative, expanding it to a more comprehensive questionnaire or scale could capture a broader range of sensations and perceptions related to naturalness.

Finally, in order to test the generalizability and the clinical relevance of the proposed approach, the next phase of the clinical trial (phases II and III) will benefit from a direct comparison with a proper control group (randomized, double-blind clinical trial).

## Use of this framework in future biomimetic neurostimulation devices

The presented neuromodulation framework based on biomimetic encoding could also be very relevant for other neuroprostheses in the CNS (e.g., Deep Brain Stimulation[83], epidural stimulation[13], ICMS[5,18]) and for bioelectronic medicine applications (e.g., vagus stimulation[84], stimulation of the autonomic nervous system[85]) having the same necessity to evoke a natural pattern of activation in a certain nervous district using artificial electrical stimulation. Indeed, the biomimetic approach has been proven to be effective for improving functional performance in other types of neural prostheses (e.g., enhanced speech intelligibility for cochlear implants[86]; improved restoration of gaze stability in vestibular prostheses[87]). We believe that an approach based on the in-silico modeling of the desired neurological function, followed by animal validation evaluated on the perceived quality of sensations and performance while doing daily tasks, will become the standard framework for the development of the novel neuroprostheses.

In future biomimetic neuro-robotic devices that restore fully natural sensations, spatial patterning can be achieved by stimulating different electrodes with spatially displaced projection fields, while temporal patterns can be elicited by temporally modulating the stimulation parameters delivered through each electrode, as proposed in our study. However, the extent to which artificially evoked neural activity must mimic that of the natural afferent inputs in order to be fully exploitable remains a critical question, especially in cases of more complex tactile features[25,88] (textures, object stiffness, shape, etc.) and proprioception.

Here, we evaluated multiple types of biomimetic patterns that were developed using the distinct response characteristics of individual afferent types. When we stimulated the entire nerve during animal experiments, the biomimetic pattern based on the aggregate afferent response (FULL biomimetic) appeared to be the most similar to its natural counterpart. Notably, when these paradigms were delivered using intraneural electrodes in humans, smaller clusters of mixed afferents were selectively activated by the different channels. Interestingly, the naturalness of the sensation, for the same encoding strategy, changed according to specific areas of the foot sole. This suggests that the imposition of the aggregate dynamics for inducing natural sensations is not optimal for every fiber cluster recruited but depends instead on the distribution of activated afferents (mechanoreceptors) and their specific role in sensory processing. We believe that neurostimulation strategies should be informed by computational modeling, which emulates realistic dynamic conditions. Moreover, we envision the usage of machine learning methods for calibrating the system[89] and predicting the most suitable stimulating pattern, together with the design of more advanced electrodes[90]. In addition to the here-presented benefit, we hypothesize that this restored natural feedback would have a positive impact on the level of incorporation of this artificial device. A more detailed assessment of this aspect should be performed through embodiment measurements[48,91].

In conclusion, our collected evidence not only amplifies the remarkable impact of biomimetic signal encoding from a scientific perspective, but it also holds immense promise in heralding the advent of the next generation of neuroprosthetic devices. New technologies, inspired by nature, have the potential to fully emulate natural neural functions lost after a disease or an injury. The possibility to naturally communicate with the brain will open new doors for science in multiple fields.

## Methods

### Modeling of all tactile afferents innervating the glabrous skin of the foot (FootSim)

In this study, we used FootSim[46], an in-silico model of the afferents innervating the footsole that simulates the neural responses to arbitrary mechanical stimuli. It is composed of two parts: (i) the mechanical part, calculating the deformation of the skin by applied stimulus and converting it into the skin stress, and (ii) firing models that generate spiking output for individual fibers of different afferent classes. Each firing model contains 11 unique parameters. The model is fitted on a dataset of tactile afferents exposed to a wide range of vibrotactile stimuli at different frequencies and amplitudes, recorded in humans using microneurography. We fitted several models for each afferent type, reflecting partially the natural response variability of different afferents observed in the empirical data.

### Design biomimetic neural stimulations using FootSim

We designed five types of biomimetic patterns based on the cumulative responses of specific afferent types. In the FootSim model, we populated the foot sole with only one type of afferents (FA1/FA2/SA1/SA2) or with a complete population of afferents (FULL biomimetic), following their realistic distribution. We applied 2 s stimuli covering the whole area of the foot. We combined ramp-and-hold stimuli (0.15 s on phase, 0.3 s off phase) with low-amplitude environmental noise (up to 0.5% of maximum amplitude of ramp-and-hold stimuli). The footSim model estimated the response of each single afferent placed on the sole of the foot. The footSim model estimates the response of every single afferent placed on the sole of the foot and provides its spiking activity. This activity for all fibers placed in the foot sole is aggregated using peri-stimulus time histogram (PSTH). We fit the smooth function resembling the PSTH and use it to modulate the frequency of stimulation. We applied the same procedure for each biomimetic paradigm. The amplitude and pulse-width of stimulation, identified during the electrode mapping procedure, were kept constant along the train. This choice was based on previous findings that frequency is the stimulation parameter more linked to evoked sensation quality[39,61,92] while charge seems to modulate intensity[93].

### Animal surgical procedure

Experiments were carried out on 2 adult cats (2–3 years old) of either sex (weighing 2.5–4.0 kg). All procedures were conducted in accordance with protocols approved by the Animal Care Committee of the Pavlov Institute of Physiology, St. Petersburg, Russia, and adhered to the European Community Council Directive (2010/63EU). The surgical procedures were similar to those in our previous studies[94,95]. The cats were deeply anesthetized with isoflurane (2–4%) delivered in $O_2$. For the induction of anesthesia, xylazine (0.5 mg/kg, i.m.) was injected. The level of anesthesia was monitored based on applying pressure to a paw (to detect limb withdrawal), as well as by checking the size and reactivity of the pupils. The trachea was cannulated, and the carotid arteries were ligated. The animals were decerebrated at the precollicular-postmammillary level to ensure pure sensory recordings without the influence of the higher structures. Access to the tibial nerve laminectomy in corresponding segments for intraspinal and DRG recording of neurons was performed (Fig. 3). A Cuff electrode (Microprobes for Life Science, Gaithersburg, MD 20879, USA) was placed after the careful dissection from surrounding tissues, around the common trunk of the tibial nerve. The exposed dorsal surface of the spinal cord was covered with warm paraffin oil. Linear shaft electrodes with 32 channels (Neuronexus, Ann Arbor, MI, USA) were carefully implanted at the spinal level L6 using stereotaxic frames. DRG

 

implant was performed by implanting the 32-channel UTAH array (Blackrock Microsystems, Salt Lake City, UT, USA) through the pneumatic injection pistol. Anesthesia was discontinued after the surgical procedures. Premedication, laminectomy, peripheral implants, decerebration, spinal implants, and DRG implants (with the preparing the positioning of the electrodes) lasted between 9 and 11 h, making the total experimental procedure 12–15 h long. During the experiment, the rectal temperature and mean blood pressure of the animals were continuously monitored and kept at $37 \pm 0.5\,°C$ and above 80 mmHg.

### Electrophysiology in decerebrated cats

Through the contact sites of the cuff electrodes, we delivered single pulses of cathodic, charge-balanced, symmetric square pulses (with a pulse width of 0.5 ms). We provided the stimulation using AM stimulators Model 2100 (A-M Systems, Sequim, WA, USA). Electromyographic and neural signals were acquired using the LTR-EU-16 recording system with LTR11 ADC (L-Card, Moscow, Russia) and the RHS recording system with 32-channel headstages (Intan Technologies, Los Angeles, CA, USA) at a sampling frequency of 25 and 30 kHz respectively. We tuned the stimulation amplitude by observing the emergence of clear sensory volleys in the dorsal spinal cord in response to low-frequency stimulation. Prior to running the experimental protocol with different stimulations, we performed tuning trials in which we inspected the effects of the peripheral stimulation on the signal in the spinal cord in real-time. We stimulated the nerve pulse by pulse (low-frequency stimulation, ~2 Hz) and increased amplitude until we achieved a robust and reproducible afferent volley (clearly observable electrical activity from the spine). These volleys carry the information from the periphery, which is then integrated and processed along the somatosensory neuroaxis. We calculate the mean (meanBaseline) and standard deviation value of the resting signal (stdBaseline). To find this threshold we calculated the mean (meanBaseline) and standard deviation value of the resting signal (stdBaseline): when peak-to-peak values of the afferent volleys were higher than meanBaseline + 2.5stdBaseline, we define that amplitude value as the amplitude of stimulation to be used. We stimulated the cat's tibial nerve with 60 μA amplitude.

We applied 5 types of biomimetic stimulation paradigms, repeating every pattern 90 times. Natural touch condition was applied by rubbing the cat's leg with a cotton swab and was repeated 15 times.

### Analysis of the animal neural data

After acquiring animal neural data, we applied all detailed analysis offline, as follows:

Pre-processing. We filtered raw signals recorded with a 32-electrode array implanted in the spinal cord, as well as signals documented with a 32-channel Utah array in the dorsal root ganglion with a comb filter to remove artifacts on 50 Hz and its harmonics. We designed a digital infinite impulse response filter as a group of notch filters that are evenly spaced at exactly 50 Hz. We removed signal drift with a high-pass 3rd-order Butterworth filter with a 30 Hz cutoff frequency. High amplitude artifacts were detected when the signal crossed a threshold equal to $15\sigma$, where we estimated background noise standard deviation[96] as $\sigma = median\,|x|\,0.6745$. Detected artifacts were zero-padded for 10 ms before and after the threshold crossing. We extracted neural signals of 2 s recorded during stimulation with every defined paradigm. Natural touch conditions produced a response of 0.8 s and we extracted for the further analysis segments in the specific trials where neural activity was robust and repeatable.

Identification of local field potential. We isolated local field potentials by band passing the neural signal between 30 and 300 Hz and averaged the signal over multiple stimuli pattern repetitions.

LFP distribution comparison. In the DRG, we extracted the channels where clear afferent volleys were visible (12 channels) and compared their overall activity in different stimulating conditions (biomimetic, tonic, and natural). We compared the distribution of the recorded LFP using the Kullback–Leibler divergence metric.

Characterization and quantification of neural spiking activity. We extracted neural spiking activity by applying a 3rd-order Butterworth digital filter to the raw signal, separating the signal in the frequency range from 800 to 5000 Hz. We detected the spikes using an unsupervised algorithm[97]. We determined the threshold value separately for each recording channel. To detect the accurate threshold value, we concatenated all data sets recorded in one place (spinal cord/DRG) that we aim to analyze in a single file. All analyzed data sets were concatenated in a single file in order to detect proper threshold values. The threshold for detection of action potentials was set to negative $3\sigma$ for signals recorded in the spinal cord and $4\sigma$ for signals recorded in the DRG, where $\sigma = median\,|x|\,0.6745$ which represents an estimation of the background standard deviation.

Multiunit activity is presented in the form of rasterplots and quantified with a peri-stimulus time histogram (PSTH). Each dot in a rasterplot represents a single detected spike, while every rasteplot row corresponds to the intra-spinal or intra-cortical activity perturbed with a single muscle nerve stimulus pulse.

When defining the proper size of the time bin, we began by plotting preliminary PSTHs with different time bin lengths and visually inspecting the results. If the bin size is too small, the histogram appears to be very noisy, with many bars that are very short and do not provide any clear information about the data's distribution. On the contrary, if the bins are too large, the histogram lacks resolution, loses important details, and does not resemble the neural dynamics well. Therefore, the bins should be large enough to smooth out the noise in the data but not so large that they oversimplify the distribution. In panel A, down the figure, we are presenting the shape of the PSTH with the optimal bin size. Moreover, we applied the Freedman–Diaconis rule for determining the histogram bin width. The optimal bin width is estimated as $2IQR/N^{1/3}$, where IQR is the interquartile range of the data and N is the number of datapoints in our dataset. By applying this transformation, the values varied from 10 to 53 ms, depending on the stimulating condition. The results of this estimating approach supported the conclusions of our visual inspection.

### Patient recruitment and surgical procedure in humans

Three unilateral transfemoral amputees were included in the study. All of them were active users of passive prosthetic devices (Ottobock 3R80) (Table S1). Ethical approval was obtained from the institutional ethics committees of the Clinical Center of Serbia (original IRB), Belgrade, Serbia (ClinicalTrials.gov identifier NCT03350061). The first patient was enrolled in the study in November 2017, and the last one in April 2018. All the participants read and signed the informed consent. During the entire duration of our study, all experiments were conducted in accordance with relevant EU guidelines and regulations. This study was performed in accordance with the principles of the Declaration of Helsinki. In this study, the outcomes reported relate to the impact of a sensitized neuroprosthetic leg on mobility (primary outcome) and cognitive effort (secondary outcome). The other outcomes on the impact of the sensitized neuroprosthetic leg are reported in previous studies as follows: Primary—mobility[7], falls avoidance, metabolic consumption[7]; Secondary—phantom limb pain (PLP)[7], embodiment, cognitive effort[7].

Four TIMEs[98] (14 active sites each) were obliquely implanted in the tibial branch of the sciatic nerve of each participant. The surgical approach used to implant TIMEs has been extensively reported elsewhere[7]. Under general anesthesia, through a skin incision over the sulcus between the biceps femoris and semitendinosus muscles, the tibial nerve was implanted with 4 TIMEs. A segment of the microelectrodes cables was drawn through 4 small skin incisions 3–5 cm higher than the pelvis ilium. The cable segments were externalized (and secured with sutures) to be available for the transcutaneous

connection with a neurostimulator. After 90 days, the microelectrodes were removed under an operating microscope in accordance with the protocol and the obtained permissions.

This study was performed within a larger set of experimental protocols aimed at assessing the impact of the restoration of sensory feedback via neural implants in three-leg amputees[7,47,48,99]. The data reported in this paper was obtained in multiple days from randomized conditions. The functional data was acquired within 2-weeks to minimize possible training effects. Given the steady performance across sessions, training effects are unlikely (Fig. S5). While Participants 1–2 participated in all experiments, because of limited time availability, Participant 3 participated only in the open-loop characterization and the assessment of naturalness while deciding not to participate in these functional tasks with biomimetics (ST and CDT). Data from the open-loop sessions testing the naturalness of the evoked sensations for Participant 3 is reported in Suppl. Fig. 3.

## Intraneural stimulation for evoking artificial sensations

Each of the TIMEs (latest generation TIME-4H) implanted in the three amputees was constituted by 14 active sites (AS) and two ground electrodes. Details concerning design and fabrication can be found in[100,101]. For each participant, 56 electrode channels were then accessible for stimulation on the tibial nerve. During the characterization procedure, the stimulation parameters (i.e., amplitude and pulse width of the stimulation train), for each electrode and AS were recorded. The electrodes were connected to an external multichannel controllable neurostimulator, the STIMEP (Axonic, and University of Montpellier)[102]. The scope of this procedure was to determine the relationships between stimulation parameters and the quality, location, and intensity of the electrically evoked sensation, as described by Petrini et al. In brief, the injected charge was linearly increased at a fixed frequency (50 Hz) and pulse-width by modulating the amplitude of the stimulation for each electrode channel. In case the stimulation range was too small for the chosen pulse width and the maximum injectable current, the pulse width was increased, and the same procedure was repeated. When the participant perceived any electrically evoked sensation, the minimum charge (i.e., perceptual threshold) was registered. The maximum charge was collected in order to avoid inducing pain or discomfort for the participant. This was repeated five times per channel and then averaged. Perceptual threshold and maximum charge were obtained for every electrode channel and have been used to choose the stimulation range. For each AS, the maximum injected charge was always below the TIME's chemical safety limit of 120 nC[103]. All the data were collected using a custom-designed psychometric platform for neuroprosthetic applications, which allowed us to collect data using standardized assessment questionnaires and scales and perform measurements over time. The psychometric platform is user-friendly and provides clinicians with all the information needed to assess the sensory feedback[104].

## Assessment of sensation naturalness

We first characterized the participants' rating of the perceived naturalness of the stimulation delivered through TIMEs in S1, S, and S3. We injected biphasic trains of current pulses lasting 2 s with an increasing phase (0.5 s), a static phase (1 s), and a decreasing phase (0.5 s) via TIMEs (Fig. 5c) using linear amplitude neuromodulation[26,39], sinusoidal pulse-width neuromodulation[10,105], Poisson frequency neuromodulation (i.e., Poisson spiking train with a mean frequency of 50 Hz, consisting in a non-biomimetic, frequency-variant stimulation, where spikes intervals are uncorrelated and exponentially distributed) and Biomimetic neurostimulation patterns constructed using FootSim (SAI-like, SAII-like, FAI-like, FAII-like, and FULL Biomimetic).

The stimulation was delivered from three ASs for S1 and S2 eliciting sensation in the Frontal met, three AS for S1 and S2 eliciting sensation in the Central met, three AS for S1 and two AS for S2 eliciting

sensation in the Lateral met, and five AS for S1 and two ASs S2 eliciting sensation in the Heel. For S3, only one AS per the four areas was tested (Fig. S3). The participants were asked to report the location (i.e., Projected Field) and naturalness, rated on a scale from 0 to 5[26,39,57]. Each condition was randomized, and each stimulation trial was repeated three times. The injected charge (amplitude and pulse width) was specific for each channel and set to the related threshold charge. Moreover, intensity ratings were also collected during each stimulation to exclude relevant intensity differences among the encoding strategies (intensity bias). For the typical time scales involved in our experiments (trials lasting on the order of minutes), neither of our participants reported relevant changes in sensation intensity, which would indicate the presence of adaptation. The specific quality descriptors of the electrically evoked sensations reported by the participants were electrode-dependent, including a multitude of sensation types (natural and unnatural)[106]. The participants were blinded to the sensory encodings used in each trial.

## Real-time biomimetic neurostimulation in a neuro-robotic leg

The neuroprosthetic system included a robotic leg with a sensorized insole with embedded pressure sensors, along with a microcontroller and a neural stimulator[102], implementing the encoding strategies and providing sensory feedback in real-time by means of implanted TIMEs. We implemented and tested: (i) no feedback (NF): the prosthesis did not provide any sensory feedback; (ii) linear amplitude neuromodulation (LIN): the prosthesis provided linear feedback from three channels of the sensorized insole (heel, lateral or medial and frontal; more details in Petrini et al.); (iii) time-discrete neuromodulation feedback (DISC): the prosthesis delivered short trains of stimulation (0.5 s) when a specific sensor was activated (heel, lateral or central and frontal) and again (0.5 s) when the load was released from that sensor (neurostimulation delivered only at the transients); (iv) biomimetic neuromodulation feedback (BIOM): the neuroprosthetic device provided the biomimetic stimulation, reported as the one eliciting more natural sensation, from three channels of the sensorized insole (heel, lateral or central and frontal). For the model-based biomimetic approach (BIOM), the corresponding frequency trains were computed previously offline by the model to reach the appropriate speed during the real-time implementation. The amplitude of the stimulation was modulated linearly with the pressure sensor output, as proposed in Valle et al. (HNM-1)[39]. In LIN and DISC, the stimulation frequency was fixed (tonic stimulation) to 50 Hz[7]. During the functional experiments reported in this work, three tactile channels (those eliciting sensation on the heel, lateral or medial, and frontal met areas) were used for sensory feedback in all the conditions. The delivered charge was similarly modulated on the three stimulating channels but in a different range. In fact, each channel was modulated between its threshold and maximum charge values identified in the last mapping session. The biomimetic stimulation patterns adopted on the three channels were selected according to the naturalness perceived per foot area (Fig. S3) in each implanted participant. In particular, FAI Biomimetic for frontal, lateral, and heel for both S1 and S2, while FULL Biomimetic neurostimulation for lateral met in both S1 and S2.

## Stairs task

During the stairs test (ST), S1 and S2 were asked to go through a course of stairs in sessions of 30 s per 10 times per condition. The setup was configured as an angular staircase endowed with six steps with a height of 10 cm and a depth of 28 cm on one side and four steps with a height of 15 cm and a depth of 27.5 cm on the other. Participants were asked to walk clockwise, climbing up the six steps and going down the four steps. Walking sessions were performed in four distinct conditions: (i) no feedback (NF); (ii) linear neuromodulation feedback (LIN); (iii) time-discrete neuromodulation feedback (DISC); (iv) biomimetic neuromodulation feedback (BIOM). All

the stimulation conditions were randomly presented to the volunteers. The gait speed for this task was reported in terms of the number of laps, as previously performed[47]. A lap is intended as going up and down the stairs and reaching the starting position again. A higher number of completed laps is indicative of a higher speed, and vice versa. S1 and S2 performed this task.

### Cognitive double task
In the cognitive double task (CDT), first S1 and S2 were instructed to walk forward for 5 m (Baseline, Fig. S4) while timing them for 10 times per 4 conditions (BIOM, LIN, DISC, and NF) performed in a random order. Subsequently, they were asked to walk for the same distance while performing a dual task (CDT). In particular, they had to spell backward in their mother-tongue language (Serbian) a five-letter word, which had not been previously presented. Also, this task was performed 10 times per 4 conditions (BIOM, LIN, DISC, and NF) performed in a random order. While the participants were performing the CDT, both the walking speed (m/s) and the accuracy of the spelling (% of correct letters) were recorded (Fig. 6b). S1 and S2 performed this task.

### Self-reported confidence
At the end of each session of ST, participants were asked to assess their self-confidence while performing the motor task, using a visual analog scale (from 0 to 10). The data were acquired in BIOM, LIN, DISC, and NF conditions in S1 and S2.

### Statistics
All data was exported and processed offline in Python (3.7.3, the Python Software Foundation), using "SciPy" and "NumPy" packages, and MATLAB (R2020a, The MathWorks, Natick, USA). All data were reported as mean values ± SD (unless otherwise indicated). The normality of data distributions was verified with a one-sample Kolmogorov–Smirnov test, using the Matlab function "kstest". The function returns a test decision for the null hypothesis that the given data comes from a standard normal distribution against the alternative that it does not come from such a distribution, using the one-sample Kolmogorov–Smirnov test. The result is 1 if the test rejects the null hypothesis at the 5% significance level or 0 otherwise. We used quantile–quantile plot (QQ-plot) for visual inspection of normality, using the Matlab function "qqplot". QQ plots quantiles of the data versus the theoretical quantile values from a normal distribution. If the distribution of the data is normal, the plot appears linear. Additionally, we checked the histogram of the data points. If the data is approximately normally distributed, the histogram should resemble a bell-shaped curve. In the case of Gaussian distribution, a one-way analysis of variance was applied, using the Matlab function "anova1". Elsewise, we performed the Kruskal–Wallis test for data that has two or more groups. A post-hoc correction was executed in case of multiple groups of data. Significance levels were 0.05 unless differently reported in the figures' captions.

### Reporting summary
Further information on research design is available in the Nature Portfolio Reporting Summary linked to this article.

## Data availability
Individual de-identified participant data and animal experimental data supporting the findings are immediately and indefinitely available at https://github.com/NatalijaKatic/Biomimetic-project.git for anyone who wishes to access the data for any purpose. Protocol for human clinical trials is given as part of the Reporting Summary. Any additional explanation of datasets or data presented in another form is available by request from the corresponding author. A translated version of the study protocol for the human clinical trial is available in the Supplementary Information file. Source data are provided in this paper.

## Code availability
Custom code used for analysis is available through Github. Code used for data collection can be made available upon request to the study PIs.

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

## Acknowledgements

The authors are deeply grateful to the three participants who freely donated months of their lives for the advancement of knowledge and for a better future for leg amputees. We thank Prof. Marco Capogrosso for their support during the animal experimentation and the related data analysis. We also thank the National Centre of Competence in Research (NCCR) Robotics for the useful collaborations. The funder had no role in the experimental design, analysis, or paper preparation or submission. All authors had complete access to data. All authors authorized the submission of the paper, but the final submission decision was made by the corresponding authors. This project has received funding from the European Research Council (ERC) under the European Union's Horizon 2020 research and innovation program (FeelAgain grant agreement no. 759998), from Gebert Ruf Stiftung (InnoBooster, MYLEG, GRS–096/21), from Swiss National Science Foundation (SNSF) (MOVE-IT no. 197271), from project IDEJE by Science Fund of the Republic of Serbia (DiabeticReTrust no. 7753949), by project (ID: 93022925/ 94030803) of the St. Petersburg State University, St. Petersburg (for N.P.), by Sirius University of Science and Technology project: NRB-RND-2115 (for P.M.). The work was carried out within the framework of the Implementation Program Priority 2030 (NUST MISIS) (for O.G.).

## Author contributions

G.V. developed the neurostimulation software for human and animal experiments, developed the biomimetic neuroprosthetic leg systems, performed the human experiments, analyzed the data collected in humans, supervised all the analyses, prepared the figures, and wrote the paper; N.K. developed the in-silico model (FootSim), generated the tested stimulation patterns using FootSim, supervised and performed the analyses on the animal data, prepared the figures and wrote the paper; D.E. performed the analyses on the animal data and prepared results related the figures; N.P. performed the animal surgical procedures. O.G. performed the animal experiments; F.M.P. developed the neuroprosthetic leg systems and performed the human experiments; P.C. and T.S. developed the TIME and delivered technical assistance for the human implantation and explanation procedures and reviewed the paper; P.M. designed and performed the animal surgical procedures and performed the animal experiments; M.B. performed the human surgeries and was responsible for all the clinical aspects of the human study; S.R. designed the study, performed and supervised the human and animal experiments, supervised the analyses, wrote the paper. All authors edited and proofread the paper.

## Competing interests

F.M.P. holds shares of "Sensars Neuroprosthetics", a start-up company dealing with the potential commercialization of neurocontrolled artificial limbs. The remaining authors declare no competing interests.
