## [Peer Review File · Nature Communications]

REVIEWER COMMENTS

Reviewer #1

The study introduced a multi-level approach aimed at enhancing movement with a prosthetic limb and minimizing paresthesia sensations in individuals with limb amputation through the development of a novel prosthesis that relays sensory feedback. This device utilized biomimetic neurostimulation patterns based on a computational model of foot innervation. To achieve more natural sensations, the research employed in-silico modeling of foot touch coding, enabling the precise definition of neural stimulation patterns that mimicked the firing of different cutaneous mechanoreceptors. These patterns encompassed single-fiber and mixed-fiber types, offering a novel way to encode mechanical skin indentation into neural stimulation policies.

A critical aspect of the study involved comparing the neural responses induced by natural touch, biomimetic neurostimulation, and tonic electrical stimulation. Results from experiments in decerebrated cats indicated that biomimetic neurostimulations resembled natural touch responses, distinguishing them from tonic stimulation. This finding was pivotal in demonstrating the potential of biomimetic approaches to recreate natural sensory experiences. Patient testing further confirmed the effectiveness of biomimetic neurostimulation strategies. Three individuals with transfemoral amputation reported experiencing more natural sensations when stimulated with biomimetic encodings compared to standard neuromodulation patterns. Incorporating biomimetic encoding into robotic prosthetic legs resulted in functional improvements. Users of these neuroprosthetic legs exhibited faster stair-walking abilities and reduced mental workload during dual-task scenarios, which holds particular significance for individuals with lower-limb amputation. The study extended its implications beyond limb amputation by suggesting the relevance of biomimetic encoding for other neuroprostheses in the central nervous system (CNS) and bioelectronic medicine applications. These technologies share the common goal of evoking natural patterns of activation within specific neural regions using artificial electrical stimulation. Computational models were used to emulate realistic dynamic conditions and the distribution of activated afferents in sensory processing. This approach informed the development of neurostimulation strategies.

In conclusion, the study underscored the substantial impact of biomimetic signal encoding on both scientific understanding and practical applications in the field of neuroprosthetics. By emulating natural neural functions lost due to disease or injury, biomimetic approaches hold great promise for enhancing the quality of life for individuals with limb amputation and advancing scientific research across various domains.

General Remarks:

The reviewed study presents a promising translational approach involving simulations, animal models, and patient studies, marking a notable advancement in lower limb neuroprosthetics. This thorough assessment covers factors like movement speed, impact of cognitive load, and subjective assessments of naturalness of evoked sensations via peripheral stimulation. However, upon closer examination of more specific execution details, the individual components appear slightly disjointed. In the upcoming review, we'll discuss these concerns while acknowledging the study's overall importance.

Firstly, the title of the paper, featuring the term "restoring," could potentially mislead readers (and possibly, hopeful patients!) regarding the study's scope. It is advisable to reconsider and possibly refine the title to accurately convey the study's objectives, as full restoration is yet to be achieved.

One noticeable omission in the study is the absence of a healthy control group for comparison. Including such a group would provide valuable context for the achieved improvements and help clarify the specific impact of the neurostimulation method.

While the study's translational design holds promise, there appears to be a gap between the theoretical framework, animal models, and its application in patients. It is not always clear how results from one level would impact later stages of the study.

There's also a question regarding the decision to present certain result figures exclusively in supplementary files. It's important to provide a clear rationale for this choice.

Lastly, the manuscript's language occasionally contains grammatical errors. Engaging a native language speaker or a professional proofreader for a review is strongly recommended.

In conclusion, while the study offers promise in improving the quality of life for individuals with limb amputation, addressing these points will refine the manuscript, ultimately enhancing its contribution to the field of neuroprosthetics and advancing scientific research in this area. In the following, more detailed suggestions are given.

Design & Statistics:

Notably, the statistical analysis falls short and requires revision, as it is currently not suitable for publication.

It is laudable that the software packages used for statistical analysis are mentioned in a separate section, please also mention the specific toolboxes and libraries used for all statistical analyses. When assessing the normality of distributions (line 759), further mention the precise test used and whether you quantified the magnitude of the deviations from normality. Please also indicate whether you verified the results using visual inspection (e.g., Q-Q plots), as statistical tests for deviation from a distribution can detect significant, but ultimately non substantial deviations, too.

ANOVA is referred to as a two-tailed test (line 760), however, while ANOVA is indeed non-directional, it is not a two-tailed test since the F-statistic is strictly positive and we only care about the right end of the distributions tail. Hence it is a one-tailed test.

Please consider reporting the exact p-values instead of just reporting the value comparison, i.e., whether the p-value is above or below your chosen threshold. Also, the reported correlation coefficients would benefit in their interpretability from confidence intervals, instead of the mere reporting of point estimates. This could for instance be achieved with bootstrapping.

Additionally, precise documentation of preprocessing pipelines, as well as analysis procedures, would greatly enhance the replicability of this study and aid future research.

There are further some concerns about inconsistency, lack of clarity, and imprecision in the reporting of statistical results and procedures, making the analysis not easily replicable. For example, some statistical tests are not explicitly mentioned and must be inferred by the reader. Please mention the specific correlation measure used (e.g. Spearman or Pearson) and indicate all correlation values with a consistent notation such as uppercase “R” or, preferably, lowercase “r” (see subchapters “The neurostimulation dynamics is transferred through somatosensory neuroaxis” and “Neural response evoked by biomimetic stimulation is more similar to the mechanically-induced activity than the one produced by tonic electrical stimulation”, where it is merely called ‘correlation coefficient’).

Also make sure that all correlation values in the above chapters are accompanied by p-values.

When assessing the similarity between distributions (lines 255-258) please indicate which method was used. If possible, also include a quantitative measure of their similarity or dissimilarity, instead of the binary decision based on p-value alone (although the figure showing the distributions is convincing for the given judgements).

When reporting differences between conditions in the patient study, it would be beneficial to introduce an estimate of the effect size, instead of a mere significance statement, as this can be useful for assessing whether improvements are meaningful. It will also be a valuable reference point for later research.

When plotting results for different subjects, please make sure that the scale of the plotted axes is the same for all subjects and avoid auto-scaling as it hampers the interpretability of the results. If plotted values do not start at 0 or ‘baseline’, consider ‘breaking’ the axis to make the viewer aware of that discontinuity. Otherwise, differences between conditions can appear deceptively large (e.g., check specifically Fig. 6). In the PSTH plots, include a horizontal bar indicating the duration corresponding to an individual bin length. This information should not be hidden in the figure description.

In case of the PSTHs of Figure 4 we are wondering about the bin count in the natural touch condition. Are some bins not reported in the diagram (assuming a bin width of 50ms and a 1s duration).

Some statements are based solely on qualitative comparisons when quantification could have been possible.

E.g., in line 285ff., conclusions about the similarity of LFPs at different channels should be, if possible, quantified and this quantification should be included in the text. The qualitative judgement, however, is well shown in Fig. S1. This also holds for lines 237-250.

The study did not include matched controls, this leaves many reported metrics of patient improvements without adequate reference points and makes them hard to interpret.

In order to interpret the results from the Cognitive double task it would be important to know how the participants' performance evolved throughout the spelling task. Did their proficiency improve across successive trials, or was there an initial habituation phase? This inquiry is of significance due to the constraints imposed by a small sample size of N=2, which precludes the application of counterbalancing conditions.

Study design:

Firstly, it is essential to note that while the study effectively and efficiently assesses aspects like naturalness of evoked sensations and impact of cognitive load, it does not encompass the full spectrum of the prosthetic user experience. Subjective aspects such as ownership, agency, effort, and fatigue should also be considered in future research to provide a more comprehensive understanding of the user experience.

If possible, provide previously reported reference values for the measures employed, both for clinical populations and healthy individuals. The inclusion of reference values would aid in contextualizing the study's findings and understanding their significance within the broader context of normal functioning.

Furthermore, the study would benefit from a more explicit rationale for the selection of specific measurement tools. While the use of a visual analog scale to assess naturalness is valid, it would be valuable to explain why alternative measures, such as embodiment questionnaires (Segil et al., 2022), were not utilized. This would provide clarity regarding the methodological choices made. If possible, provide an example of the VAS used, as well as the instructions given to the participants. A visual analog scale should only have anchors at the extreme values, yet in Fig. 5, subject 1's answers cluster mostly around integer values, which might indicate that a Likert scale, and not a VAS was used (or the VAS was understood and presented like a Likert scale).

In addition, the study would have benefitted from a quantitative analysis of gait pattern changes (gait analysis, see De Marchis et al., 2022) and does not measure effort or fatigue, which are critical aspects in evaluating the usability and long-term acceptance of neuroprosthetic devices.

To enhance the depth of the study, the incorporation of neurophysiological measures, such as EEG or EMG, and for example a within-subject comparison between healthy leg and prosthetic leg could provide valuable insights into neural adaptations and motor control differences associated with the use of neuroprostheses.

Lastly, while the assessment of naturalness using a single-item visual analog scale is informative, expanding it to a more comprehensive questionnaire or scale could capture a broader range of sensations and perceptions related to naturalness.

In conclusion, the study on novel neuroprosthetics shows promise and offers valuable insights into prosthetic user experience. Addressing these concerns and considering the suggestions made for future work would contribute to the overall robustness and comprehensiveness of the research.

Animal experiments:

The information of the animal experiments is complementary to the simulation as well as the patient study, because performing invasive electrophysiological studies at the spinal cord level in humans is not possible. However, there are concerns about the parameterization of the stimulation, as it is not optimally aligned with the previous modeling and human research. The animal experiment unequivocally demonstrates the superiority of biomimetic stimulation compared to the tonic 50Hz stimulation.

The paper lacks clarification on why the normalized mean activity of the tonic stimulation is significantly higher than for biomimetic and mechanical stimulation, as shown in Figure 4. Additionally, it is crucial to detail the methodology used for normalization to ensure transparency and reproducibility. It's important to consider the potential impact of this higher mean activity on other findings and results throughout the study.

Including the Current Source Density (CSD) correlation between the tonic and biomimetic stimulation patterns in Figure 4 D, additionally to the shown correlation between each of them and naturalistic mechanical stimulation, would enhance the paper's comprehensiveness and aid in understanding the relationships between these patterns.

The choice of using correlation to assess spatio-temporal neural dynamics between DRG and L6 should be elaborated upon in the paper, as it is not sensitive to potential temporal delays. Furthermore, if Peri-Stimulus Time Histograms (PSTH) were employed for correlation analysis (e.g., line 210f), addressing potential effects of bin width on the results is essential to evaluate the robustness of the findings. Overall, this section would benefit from a clarification of the statistical approach.

Reporting the stimulation parameters for tibial nerve stimulation in cats, as well as clarifying the definition of 'clear sensory volley', would enhance the method section's clarity and reproducibility.

The rationale for studying signal propagation in decerebrated cats, which abolishes top-down modulation, while on the other hand top-down modulation might be observed in humans using prosthetic legs, should be explicitly explained in the paper to justify this choice.

The discrepancy between simulated activity (foot sole pressure in humans, duration 2s) and actual mechanical stimulation (cat's leg with a cotton swab, duration 1s) raises concerns about confounding factors in the comparison of correlation patterns. The location of the applied mechanical, naturalistic stimulus using the cotton swab is ill-specified, since only 'cat's leg' is given as a description. Since the biomimetic stimulation is matched to the innervation of the human foot sole, it is not clear how similar this would be to touching arbitrary locations of a cat's leg. The paper should explicitly acknowledge and discuss how these differences might affect the correlation analysis and the interpretation of results. Demonstrating the equivalence of neural patterns evoked by cotton swab stimulation and stepping is crucial. If available, providing sources that support the similarity in foot and leg innervation between cats and humans would further strengthen the paper's foundation.

The decision to fix stimulation amplitude beforehand and not modulate it within the stimulation, despite previous publications suggesting its potential benefits (in hand prosthetics, Valle et al., 2018), should be discussed.

It is worth noting that the animal experiment presented an opportunity for fine-tuning the stimulation parameters to potentially achieve a more robust biomimetic activation in the spinal cord neurons. However, this valuable adjustment remained unexplored, and its absence from the report raises questions about the optimization of experimental outcomes. Providing insights into why this fine-tuning was not pursued would enhance the transparency of the research process.

Detailing the exact purpose of the animal experiment and discussing how different outcomes could have influenced the experimental protocol for the patient study is important for transparency and understanding the study's design, as well as for guiding future research.

Conclusion:

In general, the inclusion of healthy controls or of reference values, as well as the revision of the statistical analysis, could greatly improve the manuscript. Addressed issues that cannot be amended should be included in the section addressing study limitations. The study holds great potential for the development of future sensory-feedback based neuroprostheses.

Overall, addressing these concerns and incorporating could significantly enhance the clarity, rigor, comprehensiveness, and hence the impact of this paper. This feedback is provided with the intention of improving the study's quality, reproducibility, and relevance within the field. We acknowledge the immense interdisciplinary work across multiple laboratories that has contributed to this manuscript.

References:

Gait Characterization:

De Marchis, C., Ranaldi, S., Varrecchia, T., Serrao, M., Castiglia, S. F., Tatarelli, A., Ranavolo, A., Draicchio, F., Lacquaniti, F., & Conforto, S. (2022). Characterizing the Gait of People With Different Types of Amputation and Prosthetic Components Through Multimodal Measurements: A Methodological Perspective. *Frontiers in Rehabilitation Sciences*, 3. <https://www.frontiersin.org/articles/10.3389/fre.2022.804746>

Embodiment:

Segil, J. L., Roldan, L. M., & Graczyk, E. L. (2022). Measuring embodiment: A review of methods for prosthetic devices. <https://doi.org/10.3389/fnbot.2022.902162>

Combined Amplitude and Frequency Encoding:

Valle, G., Mazzoni, A., Iberite, F., D'Anna, E., Strauss, I., Granata, G., Controzzi, M., Clemente, F., Rognini, G., Cipriani, C., Stieglitz, T., Petrini, F. M., Rossini, P. M., & Micera, S. (2018). Biomimetic Intraneural Sensory Feedback Enhances Sensation Naturalness, Tactile Sensitivity, and Manual Dexterity in a Bidirectional Prosthesis. *Neuron*, 100(1), 37-45.e7. <https://doi.org/10.1016/j.neuron.2018.08.033>

Reviewer #2 (Remarks to the Author):

This manuscript describes a multi-faceted approach to the design, implementation, and evaluation of a biomimetic approach to restoring touch sensation via implanted peripheral nerve interfaces. The study is well designed and includes computational modelling, animal experiments, and trials in humans. The potential benefits of biomimetic stimulation are well known and widely accepted, so the main contribution of this work is the combined multi-faceted approach and the results of such as applied to the lower limb. The methodology is sound, although there are several areas of confusion in the data analysis that require clarification.

I have a few major questions or areas for improvement, and then several minor ones that would improve the clarity of the methodology and results.

Major -

1 - Although the content of the manuscript is clear, the quality of the writing requires improvement. I suggest using a proofing service to revise the manuscript. There are minor grammatical mistakes in most sentences, and this significantly detracts from the scientific quality.

2 - On line 318 it is clearly stated that three human participants were recruited, however line 342 refers to "both" subjects and all results from this point on only cover two participants. I could not find the rationale for only presenting data from two out of three participants - all results need to be included. Likewise, I was confused if only data from Cat 1 is presented in Figure 4 a, and if so why?

Minor -

1 - It would be good to see more discussion about the differences between upper and lower limb amputees; they have very different needs in terms of sensory feedback and this needs be discussed in order to place this work in the context of existing literature on upper limb amputees.

2 - In Fig 2 I am not clear how the FULL stimulation pattern was derived, visually it does not appear to be the sum of the others?

3 - When the cuff electrodes were implanted in the cats the manuscript states (line 197) that stimulation was increased to be slightly above threshold - what threshold?

4 - What is the rationale for using cuffs in cats and TIMEs in humans? I would expect the spatial selectivity of TIMEs to be much higher.

5 - In Fig 3 c), are the orange waveforms the mean event rate?

6 - In all figures the labels need to be improved. Please make the letters (a, b) larger and consider more consistent use of boxes to show which parts of the figure the labels apply to.

7 - When the cats' paws were stimulated with a cotton bud how did you synchronize this with the neural recording? Fig 4 a) implies a mechanical stimulus that was acting down on the paw and not a cotton bud. Why is the spiking activity shown in respect of "stimuli length" and not time?

8 - In the human trials, the order of the tasks was randomised. However, there will still have been some learning as the participants repeated the task - how did you account for this?

9 - I was surprised that the self-reported confidence did not show as significant an improvement as the speed of the task. Do you know why this was?

10 - Lines 486/487, please do not use "him" when referring to participants unless you are specifically referring to a male participant. Please use "them/they" instead.

We thank the reviewers for the appreciation of our work and for very important and useful comments that helped us improve the quality of our manuscript. Our point-to-point replies (in blue, together with the modified text in the manuscript, in *italic*) to reviewers' comments (in black) are provided below. We also performed additional checks, corrected the typos and proof-read the manuscript by a native speaker (the most substantial changes are in *light-blue italic*).

Reviewer #1

General Remarks:

The reviewed study presents a promising translational approach involving simulations, animal models, and patient studies, marking a notable advancement in lower limb neuroprosthetics. This thorough assessment covers factors like movement speed, impact of cognitive load, and subjective assessments of naturalness of evoked sensations via peripheral stimulation. However, upon closer examination of more specific execution details, the individual components appear slightly disjointed. In the upcoming review, we'll discuss these concerns while acknowledging the study's overall importance.

We thank the reviewer for highlighting the importance of our study and for all suggestions. In the following letter we address the comments of the reviewer, by point-to-point replies.

Firstly, the title of the paper, featuring the term "restoring," could potentially mislead readers (and possibly, hopeful patients!) regarding the study's scope. It is advisable to reconsider and possibly refine the title to accurately convey the study's objectives, as full restoration is yet to be achieved.

We thank the reviewer for this suggestion, and agree. We changed the title to "Biomimetic computer-to-brain communication *enhancing* naturalistic touch sensations via peripheral nerve stimulation".

One noticeable omission in the study is the absence of a healthy control group for comparison. Including such a group would provide valuable context for the achieved improvements and help clarify the specific impact of the neurostimulation method.

Since a direct comparison in the same exact tasks (angular stairs walking and our dual task overground walking+spelling letters) is not possible, we compared the general improvement in walking of healthy people with transfemoral amputation, while using a similar assistive device, without biomimetic feedback. Commercial microprocessor-controlled knees improve participants' self-selected walking speed by about 8% compared to mechanically passive devices (Orendurff, M. S. et al. Gait efficiency using the C-Leg. *J. Rehabil. Res. Dev.* 43, 239–246 (2006)). In this study, we show that the speed of participants in stair tasks while using a microprocessor-controlled knee (RHEO KNEE XC) was improved even more by biomimetic sensory feedback (>13% in S1 and >22% in S2). Regarding the self-confidence results, many studies have reported the decreased self-confidence in walking for lower-limb amputees and its strong relationship with mobility and walking abilities (Miller et al., 2001 APMR, Miller et al., 2011 Prosthetics and Orthotics International and Sions JM, et al., 2020 Physiother Theory Pract.). Also previous studies with upper-limb amputees demonstrated that sensory feedback increased subjects' confidence (i.e., self-efficacy) and was directly correlated with prosthesis accuracy in functional tasks (Schiefer et al., 2018 PlosOne, Graczyk et al., 2019 PlosOne). In that case the increments between the use of the prosthesis with and without tactile feedback were between +40% and +60% in object identification. In our case, the improvements of using the prosthesis with and without biomimetic feedback were similar +46% and +58% for S1 and S2 respectively.). As suggested, to contextualize our results, we included in the discussion:

"Commercial microprocessor-controlled knees improve participants' self-selected walking speed by about 8% compared to mechanically passive devices (Orendurff, M. S. et al. Gait efficiency using the C-Leg. J. Rehabil. Res. Dev. (2006)). In this study, we showed that the speed of participants in stair tasks while using a microprocessor-controlled knee (RHEO KNEE XC) was improved even more by biomimetic sensory feedback (>13% in S1 and >22% in S2). Regarding the self-confidence results, many studies have reported the decreased self-confidence in walking for lower-limb amputees and its strong relationship with mobility and walking abilities (Miller et al., 2001 APMR, Miller et al., 2011 Prosthetics and Orthotics International and Sions JM, et al., 2020 Physiother Theory Pract.). Also previous studies with upper-limb amputees have then shown that sensory feedback increased subjects' confidence (i.e., self-efficacy) and was directly correlated with prosthesis accuracy in functional tasks (Schiefer et al., 2018 PlosOne, Graczyk et al., 2019 PlosOne)."

In addition, a proper control group for implantable medical devices tests is a matter of an ongoing discussion in the neuroprosthetic field. Indeed, an effective and ethical condition as control is not trivial since implanting healthy individuals (or disabled and then non using the device) is not a recommendable option. Therefore cross-over intrasubject studies are possibly the most commendable. Moreover, for a similar type of pilot clinical trial aiming to demonstrate a proof-of concept of the technology, the commonly adopted design is the cross-over clinical trial with 'within-group comparison' as our study. Indeed, it has also been done for: upper-limb peripheral

stimulation (Graczyk et al., 2016 Science Trans Med, Catalan et al 2020 JNEM), lower-limb peripheral stimulation (Kim, D., et al (2023). *Science robotics*), spinal cord stimulation (Wagner et al., 2018 Nature), intracortical BCI (Flesher et al., 2021 Science), iFES (Ajiboye et al., 2017 Lancet) or speech BCI decoding (Willet 2023 Cell).

In our study we followed the same principle: the interventional model adopted is a single group assignment. This method is a clinical study design in which there is only one intervention arm available: all trial participants receive the same intervention (including dosage, frequency, and delivery method) being studied for the duration of the trial. In our case, the controls were: 1) the no feedback stimulation (used as sham or baseline condition) and 2) the different encoding stimulations (Continuous Stimulation and Discrete Stimulation). Each condition was counterbalanced and randomized in each test. The participants were blind to the stimulation condition.

However, we agree with the reviewer that a direct comparison with a control age and mobility-matched group could be useful to generalize the results, and also for a better view of the clinical implications of this intervention, in a future, bigger cohort of patients. Therefore, we added in the discussion:

“Finally, in order to test the generalizability and the clinical relevance of the proposed approach, the next phase of the clinical trial (phase II and III) will benefit from a direct comparison with a proper control group (randomized, double-blind clinical trial).”

While the study's translational design holds promise, there appears to be a gap between the theoretical framework, animal models, and its application in patients. It is not always clear how results from one level would impact later stages of the study.

We apologize for the lack of clarity. To this aim, we added an introductory paragraph in the Results section to better explain the flow of the study: *“To design an optimal neurostimulation strategy, based on a bio-inspired computation, able to effectively convey somatosensation, we exploited a trifold framework including computational modeling, animal testing and human clinical trial (Fig.1). Computational modelling consisted of designing of neurostimulation strategies to mimic the natural touch computation, shaped by a realistic in-silico model, called FootSim48. This model allows us to emulate the spatio-temporal dynamics of the natural touch code considering all the tactile afferents innervating the plantar area of the foot. Experimental steps involved tests in animals where we showed that the biomimetic paradigm was transmitted through the somatosensory neuroaxis following the same neural dynamics from the periphery to the spinal cord. Neural responses evoked at multiple levels of the somatosensory neuroaxis, showed higher similarity to the neural activity induced by natural touch compared to the standard stimulation methods, suggesting that these biomimetic patterns could potentially be the optimal encoding strategy for human neuroprosthetics. Indeed, we implemented the sensory encodings in a closed-loop neuroprosthetic leg, comparing both sensation naturalness and feedback performance in the context of everyday life activities.”*

There's also a question regarding the decision to present certain result figures exclusively in supplementary files. It's important to provide a clear rationale for this choice.

Due to the limited number of panels and figures allowed by nature communication in the main text, we placed some figures in the supplement doc to allow the reader to still visualize the data presented. In particular, we move to supplementary material:

- 1) LFPs measured at the spinal level for all the biomimetic stimulation conditions, in which the most relevant difference was reported in the main figure (FULL/50Hz and Natural).
- 2) Data from Subject 3, who participated only in open loop tests.
- 3) Breakdown of the sensation naturalness per foot area, while the main (cumulative) part of it is in Fig.5
- 4) Baseline control condition in CDT for both subjects.
- 5) Patients' details. Given the amount of information and data presented in the manuscript, we decided to move these minor results in the supplement.

Lastly, the manuscript's language occasionally contains grammatical errors. Engaging a native language speaker or a professional proofreader for a review is strongly recommended.

We thank the reviewer for pointing it out. As suggested, we performed the proof-reading by a native English corrector.

In conclusion, while the study offers promise in improving the quality of life for individuals with limb amputation, addressing these points will refine the manuscript, ultimately enhancing its contribution to the field of neuroprosthetics and advancing scientific research in this area. In the following, more detailed suggestions are given.

We thank the reviewer for such a detailed and helpful review, and for giving us encouragement for improving both the manuscript and future studies.

Design & Statistics:

It is laudable that the software packages used for statistical analysis are mentioned in a separate section, please also mention the specific toolboxes and libraries used for all statistical analyses. When assessing the normality of distributions (line 759), further mention the precise test used and whether you quantified the magnitude of the deviations from normality. Please also indicate whether you verified the results using visual inspection (e.g., Q-Q plots), as statistical tests for deviation from a distribution can detect significant, but ultimately non substantial deviations, too.

We thank the reviewer for this suggestion. We have expanded the Statistics section of Methods, changing from: “All data were exported and processed offline in Python (3.7.3, the Python Software Foundation) and MATLAB (R2020a, The MathWorks, Natick, USA). All data were reported as mean values \pm SD (unless otherwise indicated). The normality of data distributions was verified. In case of Gaussian distribution, two-tailed analysis of variance (ANOVA) test was applied. Elsewise, we performed the Wilcoxon rank-sum test. Post-hoc correction was executed in case of multiple groups of data. Significance levels were 0.05 unless differently reported in the figures’ captions. In the captions of the figures, we reported the used statistical tests for each analysis and its result, along with the number of repetitions (n) and p values for each experiment. “

To:

“All data was exported and processed offline in Python (3.7.3, the Python Software Foundation), using “SciPy” and “NumPy” packages, and MATLAB (R2020a, The MathWorks, Natick, USA). All data were reported as mean values \pm SD (unless otherwise indicated). The normality of data distributions was verified with a one-sample Kolmogorov-Smirnov test, using Matlab function “kstest”. The function returns a test decision for the null hypothesis that the given data comes from a standard normal distribution, against the alternative that it does not come from such a distribution, using the one-sample Kolmogorov-Smirnov test. The result is 1 if the test rejects the null hypothesis at the 5% significance level, or 0 otherwise. We used quantile-quantile plot (QQ-plot) for visual inspection of normality, using Matlab function “qqplot”. QQ plot quantiles of the data versus the theoretical quantile values from a normal distribution. If the distribution of the data is normal, the plot appears linear. Additionally, we checked the histogram of the data points. If the data is approximately normally distributed, the histogram should resemble a bell-shaped curve. In case of Gaussian distribution, one-way analysis of variance (ANOVA) was applied, using Matlab function “anova1”. Elsewise, we performed Kruskal-Wallis test for data that has two or more groups. Post-hoc correction was executed in case of multiple groups of data. Significance levels were 0.05 unless differently reported in the figures’ captions. “

Here we are providing one example of the statistical procedure we applied on all data:

- Verify the normality of data distributions with one-sample Kolmogorov-Smirnov test (using Matlab functions)
[h, p] = kstest (data);
h=1 -> the test rejects the null hypothesis at the 5% significance level
p=1.2271e-10 -> the rejection is very strong
- Verifying the results with visual inspection (QQ-plot) using Matlab function *qqplot*

- Verifying the results with visual inspection (histogram) using Matlab function *hist*:

ANOVA is referred to as a two-tailed test (line 760), however, while ANOVA is indeed nondirectional, it is not a two-tailed test since the F-statistic is strictly positive and we only care about the right end of the distributions tail. Hence it is a one-tailed test.

We are very sorry for this misunderstanding; the reviewer is absolutely right. The information is corrected in the manuscript, we performed a one-way analysis of variance using Matlab function "anova1".

Please consider reporting the exact p-values instead of just reporting the value comparison, i.e., whether the p-value is above or below your chosen threshold. Also, the reported correlation coefficients would benefit in their interpretability from confidence intervals, instead of the mere reporting of point estimates. This could for instance be achieved with bootstrapping.

We thank the reviewer for advice. We modified along the whole manuscript, by adding exact p-values and confidence intervals for correlations. In particular, we have modified and introduced the new values in the whole results section for both animal and human experiments (Figure 3,4,5,6).

Only for p values less than .001, we report them as $p < .001$, instead of the actual exact p value. Expressing p to more than 3 significant digits does not add useful information since precise p values with extreme results are sensitive to biases or departures from the statistical model.

Additionally, precise documentation of preprocessing pipelines, as well as analysis procedures, would greatly enhance the replicability of this study and aid future research.

We expanded the paragraphs in the Methods section as well as throughout the manuscript with the more detailed procedures. Model used for creating biomimetic paradigms is deposited on GitHub at <https://github.com/ActiveTouchLab/footsim-python> and is publicly available. Code for the analysis of data resulting from animal experiments can be found on GitHub at <https://github.com/NatalijaKatic/Biomimetic.git>.

There are further some concerns about inconsistency, lack of clarity, and imprecision in the reporting of statistical results and procedures, making the analysis not easily replicable. For example, some statistical tests are not explicitly mentioned and must be inferred by the reader. Please mention the specific correlation measure used (e.g. Spearman or Pearson) and indicate all correlation values with a consistent notation such as uppercase "R" or, preferably, lowercase "r" (see subchapters "The neurostimulation dynamics is transferred through somatosensory neuroaxis" and "Neural response evoked by biomimetic stimulation is more similar to the mechanically-induced activity than the one produced by tonic electrical stimulation", where it is merely called 'correlation coefficient'). Also make sure that all correlation values in the above chapters are accompanied by p-values.

We implemented all the corrections suggested by the reviewer regarding the statistics: statistical tests types, specific correlation measures and p values for correlations.

When assessing the similarity between distributions (lines 255-258) please indicate which method was used. If possible, also include a quantitative measure of their similarity or dissimilarity, instead of the binary decision based on p-value alone (although the figure showing the distributions is convincing for the given judgements).

We thank the reviewer for the suggestion and we apologize for the lack of precision. We compared the dissimilarity of LFP amplitude distributions using Kruskal Wallis statistical test and similarity of it using Kolmogorov-Smirnov test. None of the distributions showed statistical similarity. Tonic stimulation and natural touch responses showed statistically different amplitude distribution ($p < 0.001$). The same conclusion arrived when we compared tonic and biomimetic stimulation ($p < 0.001$), while response on biomimetic stimulation didn't show statistical difference with the natural condition ($p = 0.68$, significance level 1%, chi-square value 37×10^4).

This indicates that by stimulating the nerve with biomimetic stimulation we are resembling better the natural neural response than when stimulating it with constant stimulation, however we are still not achieving very high similarity of neural responses. We expanded the Results section of the manuscript as follows:

“We extracted the DRG most active channels where clear LFPs were visible and investigated their amplitude variations. More in detail, we compared the amplitude distribution of recorded LFP using Kruskal Wallis test for testing the dissimilarity and two-sample Kolmogorov-Smirnov test for testing similarity of distributions (Fig.4B, see Methods). Tonic stimulation and natural touch responses showed statistically different amplitude distribution ($p < 0.001$). The same conclusion arrived when we compared tonic and biomimetic stimulation ($p < 0.001$), while response on biomimetic stimulation didn't show statistical difference with the natural condition ($p = 0.68$, significance level 1%, chi-square value 37×10^4). None of the distributions showed statistical similarity. As an addition, we tested natural touch conditions in one more cat to investigate the cross-subject similarities of neural dynamics. The distribution of LFP amplitude is showing the same trend as in cat 2, however, statistical similarity is not confirmed. This evidence suggests that the naturally evoked response follows a specific, potentially generalizable trend, rather than being completely individual.”

We have also added the paragraph in the Methods section:

“LFP distribution comparison, we have compared the distribution of the recorded LFP using Kruskal-Wallis statistical test for testing dissimilarity and two-sample Kolmogorov-Smirnov test for testing similarity between two distributions. In the DRG we extracted the channels where clear afferent volleys were visible (12 channels) and compared their overall activity in different stimulating conditions (biomimetic, tonic and natural).”

When reporting differences between conditions in the patient study, it would be beneficial to introduce an estimate of the effect size, instead of a mere significance statement, as this can be useful for assessing whether improvements are meaningful. It will also be a valuable reference point for later research.

We thank the reviewer for this comment. We added effect size for the patient results (The f effect size statistic, is computed by G*Power, as the standardized average dispersion among the group means).

We have modified and introduced the new values in the results section for the human experiments (Figure 5 and 6).

When plotting results for different subjects, please make sure that the scale of the plotted axes is the same for all subjects and avoid auto-scaling as it hampers the interpretability of the results. If plotted values do not start at 0 or 'baseline', consider 'breaking' the axis to make the viewer aware of that discontinuity. Otherwise, differences between conditions can appear deceptively large (e.g., check specifically Fig. 6).

We modified Fig. 6 as suggested:

We implemented the same changes also in the Supp Figure 5.

In the PSTH plots, include a horizontal bar indicating the duration corresponding to an individual bin length. This information should not be hidden in the figure description. In case of the PSTHs of Figure 4 we are wondering about the bin count in the natural touch condition. Are some bins not reported in the diagram (assuming a bin width of 50ms and a 1s duration).

We thank the reviewer for the comment, and we apologize for it. It was our mistake in the Methods section; the natural condition produced a response of 0.8s. We corrected the Pre-processing paragraph in the Methods section, from:

“Natural touch conditions produced a response of 1 s and the signal where neural activity was observable was extracted.”

To: “Natural touch conditions produced a response of 0.8 s and we extracted for the further analysis segments in the specific trials where neural activity was robust and repeatable.”

Notation regarding the bin length has been added on the figure.

Some statements are based solely on qualitative comparisons when quantification could have been possible. E.g., in line 285ff., conclusions about the similarity of LFPs at different channels should be, if possible, quantified and this quantification should be included in the text. The qualitative judgement, however, is well shown in Fig. S1. This also holds for lines 237-250.

We thank the reviewer for the suggestion. We have changed the paragraph from:

“We compared the correlation between the LFP in the first channel of the intraspinal array and all the other channels (Fig. S1). In the natural touch condition, similarity between the neural activity is high in the first few channels and it is diminished when looking at more ventral recordings, in both animals. When the nerve was electrically stimulated, similarity between neural activity recorded with the different channels through the spinal

array is high. The biomimetic neurostimulation elicited a less similarity along the spinal axes than tonic stimulation. Full population biomimetic pattern showed to be the more promising one compared to the paradigms created by mimicking response of specific afferent types. Despite being significantly different from the natural touch, biomimetic stimulation based on aggregate population of afferent responses shares a striking similarity with it, setting it significantly apart from the tonic, 50 Hz stimulation.”

To:

“We compared the Pearson correlation coefficients between the LFP in the first channel of intraspinal array and all the other channels (Fig. S1). In the natural touch condition, similarity between the neural activity is high in the first few channels (2nd channel correlation coefficient is 0.37, 3rd 0.34, 4th 0.1; $p < 0.001$, $\alpha = 0.05$) and it is diminished when looking at more ventral recordings (less than 0.1 correlation coefficient, leading to median correlation coefficient value of 0.04, 25th percentile 0.02 and 75th percentile 0.05). When a nerve was electrically stimulated, similarity between neural activity recorded with the different channels through spinal array is high (FA1: median 0.94, 25th perc. 0.90, 75th perc. 0.95; FA2: median 0.86, 25th perc. 0.80, 75th perc. 0.92; SA1: median 0.87, 25th perc. 0.83, 75th perc. 0.89; SA2: median 0.92, 25th perc. 0.88, 75th perc. 0.93; Fig. S1). The biomimetic neurostimulation elicited less similarity along the spinal axes than tonic stimulation (biomimetic FULL: median 0.6, 25th perc. 0.45, 75th perc. 0.67; 50Hz tonic: median 0.88, 25th perc. 0.83, 75th perc. 0.9; Fig. S1). Full population biomimetic pattern showed to be the more promising one compared to the paradigms created by mimicking response of specific afferent types. Despite being significantly different from the natural touch, biomimetic stimulation based on aggregate population of afferent responses shares a striking similarity with it, setting it significantly apart from the tonic, 50 Hz stimulation (p values: FA1- FULL biom: < 0.001 ; FA2- FULL biom: 0.001 ; SA1- FULL biom: 0.01 ; SA2- FULL biom: < 0.001 ; 50Hz-FULL biom: 0.001 ; Natural- FULL biom: 0.035 ; 50Hz-Natural: $p < 0.001$; significance level 1%, chi-square value 187.4).”

We have also added the quantification of p values in the figure caption.

The study did not include matched controls, this leaves many reported metrics of patient improvements without adequate reference points and makes them hard to interpret.

Since a direct comparison in the same exact tasks (angular stairs walking and our dual task overground walking+spelling letters) is not possible we compared the general improvement in walking in people with transfemoral amputation, with the use of an assistive device. Commercial microprocessor-controlled knees improve participants’ self-selected walking speed by about 8% compared to mechanically passive devices (Orendurff, M. S. et al. Gait efficiency using the C-Leg. *J. Rehabil. Res. Dev.* 43, 239–246 (2006)). In this study, we show that the speed of participants in stair tasks while using a microprocessor-controlled knee (RHEO KNEE XC) was improved even more by biomimetic sensory feedback ($>13\%$ in S1 and $>22\%$ in S2). Regarding the self-confidence results, many studies have reported the decreased self-confidence in walking for lower-limb amputees and its strong relationship with mobility and walking abilities (Miller et al., 2001 APMR, Miller et al., 2011 Prosthetics and Orthotics International and Sions JM, et al., 2020 Physiother Theory Pract.). Also, previous studies with upper-limb amputees have then shown that sensory feedback increased subjects’ confidence (i.e., self-efficacy) and was directly correlated with prosthesis accuracy in functional tasks (Schiefer et al., 2018 PlosOne, Graczyk et al., 2019 PlosOne). In that case the increments between the use of the prosthesis with and without tactile feedback were between $+40\%$ and $+60\%$ in object identification. In our case, the improvements of using the prosthesis with and without biomimetic feedback were similar $+46\%$ and $+58\%$ for S1 and S2 respectively. As suggested, to contextualize our results, we included in the discussion:

*“Commercial microprocessor-controlled knees improve participants’ self-selected walking speed by about 8% compared to mechanically passive devices (Orendurff, M. S. et al. Gait efficiency using the C-Leg. *J. Rehabil. Res. Dev.* (2006)). In this study, we showed that the speed of participants in stair tasks while using a microprocessor-controlled knee (RHEO KNEE XC) was improved even more by biomimetic sensory feedback ($>13\%$ in S1 and $>22\%$ in S2). Regarding the self-confidence results, many studies have reported the decreased self-confidence in walking for lower-limb amputees and its strong relationship with mobility and walking abilities (Miller et al., 2001 APMR, Miller et al., 2011 Prosthetics and Orthotics International and Sions JM, et al., 2020 Physiother Theory Pract.). Importantly, previous studies on upper-limb amputees have then shown that sensory feedback increased subjects’ confidence (i.e., self-efficacy) and was directly correlated with prosthesis accuracy in functional tasks (Schiefer et al., 2018 PlosOne, Graczyk et al., 2019 PlosOne).”*

As answered in detail above, while we agree with the reviewer that a direct comparison with a control, age and mobility-matched group, could be useful to generalize the results and for a better view of the clinical implications of this intervention, here it was impracticable for above-mentioned reasons. Yet, we added in the discussion: *“In order to test the generalizability and the clinical relevance of the proposed approach, the next phase of the clinical trial (phase II and III) will benefit from a direct comparison with a proper control group (randomized, double-blind clinical trial).”*

In order to interpret the results from the Cognitive double task it would be important to know how the participants' performance evolved throughout the spelling task. Did their proficiency improve across successive trials, or was there an initial habituation phase? This inquiry is of significance due to the constraints imposed by a small sample size of $N=2$, which precludes the application of counterbalancing conditions.

In order to address this question, we evaluated the spelling accuracy performance of both subjects across sessions in our single group assignment.

Fig S6

Both subjects showed low Spearman's rank correlation coefficients (between 0.03 and 0.5) across sessions, suggesting almost no learning or initial habituation effect on their spelling performance.

It is important to consider that the conditions are counterbalanced and randomized in each task, avoiding a direct bias due to the learning effect in a specific condition.

Thanks to this reviewer's comment, we decided to add a new supplementary figure showing these new results (Fig. S5). In the results, we added:

"Analyzing the spelling accuracy, both subjects showed low Spearman's rank correlation coefficients (between 0.03 and 0.5) across sessions, suggesting no learning or initial habituation effect on their performance (Fig.S5)."

Study design:

Firstly, it is essential to note that while the study effectively and efficiently assesses aspects like naturalness of evoked sensations and impact of cognitive load, it does not encompass the full spectrum of the prosthetic user experience. Subjective aspects such as ownership, agency, effort, and fatigue should also be considered in future research to provide a more comprehensive understanding of the user experience. If possible, provide previously reported reference values for the measures employed, both for clinical populations and healthy individuals. The inclusion of reference values would aid in contextualizing the study's findings and understanding their significance within the broader context of normal functioning.

We thank the reviewer for this advice. We included these points in the discussion:

"To complete the assessment of this biomimetic neuroprosthesis, future clinical trials should include quantitative analyses of gait pattern changes (De Marchis et al., 2022, Valle et al., 2021 Sci Adv) fatigue and neurophysiological measures, which are critical aspects for long-term acceptance of these devices. Moreover,

while the assessment of naturalness using a single-item VAS is informative, expanding it to a more comprehensive questionnaire or scale could capture a broader range of sensations and perceptions related to naturalness.”

We added in the discussion:

“In addition to the here-presented benefit, we hypothesize that this restored natural feedback would have a positive impact on the level of incorporation of this artificial device, indeed a more detailed assessment of this aspect should be performed through embodiment measurements (Segil et al., 2022, Preatoni 2021 Curr Bio)”

Notably, it is worth noting that the primary goal of the study relates to the design of somatosensory encoding via electrical stimulation of the peripheral nerves.

Regarding the reference of healthy individuals, the interventional model adopted in our trial follows the single group assignment. For a similar type of pilot clinical trial aiming to demonstrate a proof-of concept of the technology, the commonly adopted design is the cross-over clinical trial with ‘within-group comparison’ as our study. Indeed, it has also been done for: upper-limb peripheral stimulation (Graczyk et al., 2016 Science Trans Med, Catalan et al 2020 JNEM), lower-limb peripheral stimulation (Kim, D., et al (2023). *Science robotics*), spinal cord stimulation (Wagner et al., 2018 Nature), intracortical BCI (Flesher et al., 2021 Science), iFES (Ajiboye et al., 2017 Lancet) or speech BCI decoding (Willet 2023 Cell). This method is a study design in which there is only one intervention arm available to all study participants. In other words, all trial participants receive the same intervention (including dosage, frequency, and delivery method) being studied for the duration of the trial. In our case, the controls were: 1) the no feedback stimulation (used as sham or baseline condition) and 2) the different encoding stimulations (Continuous Stimulation and Discrete Stimulation). Each condition was counterbalanced and randomized in each test. The participants were blind to the stimulation condition.

Since a direct comparison in the same exact tasks (angular stairs walking and dual task overground walking+spelling letters) is not possible we compared the general improvement in walking in healthy people with transfemoral amputation, with the use of an assistive device. Commercial microprocessor-controlled knees improve participants’ self-selected walking speed by about 8% compared to mechanically passive devices (Orendurff, M. S. et al. Gait efficiency using the C-Leg. *J. Rehabil. Res. Dev.* 43, 239–246 (2006)). In this study, we show that the speed of participants in stair tasks while using a microprocessor-controlled knee (RHEO KNEE XC) was improved even more by biomimetic sensory feedback (>13% in S1 and >22% in S2). Regarding the self-confidence results, many studies have reported the decreased self-confidence in walking for lower-limb amputees and its strong relationship with mobility and walking abilities (Miller et al., 2001 APMR, Miller et al., 2011 Prosthetics and Orthotics International and Sions JM, et al., 2020 Physiother Theory Pract.). Notably, previous studies on upper-limb amputees have then shown that sensory feedback increased subjects’ confidence (i.e., self-efficacy) and was directly correlated with prosthesis accuracy in functional tasks (Schiefer et al., 2018 PlosOne, Graczyk et al., 2019 PlosOne). In that case the increments between the use of the prosthesis with and without tactile feedback were between +40% and +60% in object identification. In our case, the improvements of using the prosthesis with and without biomimetic feedback were similar +46% and +58% for S1 and S2 respectively. As suggested, to contextualize our results, we included in the discussion:

“Commercial microprocessor-controlled knees improve participants’ self-selected walking speed by about 8% compared to mechanically passive devices (Orendurff, M. S. et al. Gait efficiency using the C-Leg. J. Rehabil. Res. Dev. (2006)). In this study, we showed that the speed of participants in stair tasks while using a microprocessor-controlled knee (RHEO KNEE XC) was improved even more by biomimetic sensory feedback (>13% in S1 and >22% in S2). Regarding the self-confidence results, many studies have reported the decreased self-confidence in walking for lower-limb amputees and its strong relationship with mobility and walking abilities (Miller et al., 2001 APMR, Miller et al., 2011 Prosthetics and Orthotics International and Sions JM, et al., 2020 Physiother Theory Pract.). Importantly, previous studies on upper-limb amputees have then shown that sensory feedback increased subjects’ confidence (i.e., self-efficacy) and was directly correlated with prosthesis accuracy in functional tasks (Schiefer et al., 2018 PlosOne, Graczyk et al., 2019 PlosOne).”

Regarding the naturalness of the direct neural stimulation, low values (around 2/5) were also reported in upper-limb amputees when using linear and non-biomimetic approaches (Valle et al., 2018 Neuron).

Furthermore, the study would benefit from a more explicit rationale for the selection of specific measurement tools. While the use of a visual analog scale to assess naturalness is valid, it would be valuable to explain why alternative measures, such as embodiment questionnaires (Segil et al., 2022), were not utilized. This would provide clarity regarding the methodological choices made. If possible, provide an example of the VAS used, as well as the instructions given to the participants. A visual analog scale should only have anchors at the extreme values, yet in Fig. 5, subject 1’s answers cluster mostly around integer values, which might indicate that a Likert scale, and not a VAS was used (or the VAS was understood and presented like a Likert scale).

We apologize for the lack of clarity. We adopted this specific type of assessment for multiple reasons. It is worth mentioning that a standard protocol of assessment for artificial sensory feedback is not defined yet. We

measured evoked sensation naturalness using an absolute scale (VAS) to quantify how far this artificial sensation would be from natural touch, in fact one of the critics of the Valle et al., 2018 Neuron, was about the adopted method to compare the different encoding strategies: this method measured the relative sensation naturalness since each answer is based on the direct comparison with another encoding. Here, instead the use of absolute scales allowed for a more general evaluation in respect to the natural counterpart (also used in Valle et al, 2018 Sci rep, Graczyk et al., 2018 Sci Rep).

Finally, the dual task, also related to everyday life, consisted in a combination of motor and cognitive skills measuring an integration of restored sensory information in patients' motor schema. Nevertheless, we agree with the reviewer that more detailed assessment could be performed in the next phase of experiments to better quantify the cognitive integration of this more natural feedback using embodiment measurements.

We added in the discussion:

"In addition to the here-presented benefit, we hypothesize that this restored natural feedback would have a positive impact on the level of incorporation of this artificial device, indeed a more detailed assessment of this aspect should be performed through embodiment measurements (Segil et al., 2022, Preatoni 2021 Curr Bio)"

Regarding the adopted scale, a VAS scale was presented to the subjects, but we presented the data in a Likert scale. We added this information in the Figure 5 caption:

"(d) Naturalness ratings (VAS scale, displayed on a scale 0-5) of the perceived sensation elicited exploiting different stimulation strategies in two subjects. Insets: Group comparison between linear vs biomimetic stimulations."

In addition, the study would have benefitted from a quantitative analysis of gait pattern changes (gait analysis, see De Marchis et al., 2022) and does not measure effort or fatigue, which are critical aspects in evaluating the usability and long-term acceptance of neuroprosthetic devices. To enhance the depth of the study, the incorporation of neurophysiological measures, such as EEG or EMG, and for example a within-subject comparison between healthy leg and prosthetic leg could provide valuable insights into neural adaptations and motor control differences associated with the use of neuroprostheses. Lastly, while the assessment of naturalness using a single-item visual analog scale is informative, expanding it to a more comprehensive questionnaire or scale could capture a broader range of sensations and perceptions related to naturalness.

We thank the reviewer for this advice. We included these points in the discussion.

"To complete the assessment of this biomimetic neuroprosthesis, next clinical trials should benefit from including quantitative analyses of gait pattern changes (De Marchis et al., 2022, Valle et al., 2021 Sci Adv) fatigue and neurophysiological measures, which are critical aspects for long-term acceptance of these devices. Moreover, while the assessment of naturalness using a single-item VAS is informative, expanding it to a more comprehensive questionnaire or scale could capture a broader range of sensations and perceptions related to naturalness."

Nevertheless, confidence and mobility (macroscopic parameters in rehab) result to be among the clearest and simplest parameters showing the impact of sensory feedback on gait. Therefore, we suggest considering these important features as the global evidence of a better sensorimotor strategy adopted by the subject. This could be of great importance in the design and the evaluation of the benefits of new somatosensory neuroprostheses (macroscopic versus microscopic benefits).

In conclusion, the study on novel neuroprosthetics shows promise and offers valuable insights into prosthetic user experience. Addressing these concerns and considering the suggestions made for future work would contribute to the overall robustness and comprehensiveness of the research.

Thanks for this comment and help for improving the manuscript's overall quality.

Animal experiments:

The information of the animal experiments is complementary to the simulation as well as the patient study, because performing invasive electrophysiological studies at the spinal cord level in humans is not possible. However, there are concerns about the parameterization of the stimulation, as it is not optimally aligned with the previous modeling and human research. The animal experiment unequivocally demonstrates the superiority of biomimetic stimulation compared to the tonic 50Hz stimulation.

We thank the reviewer for the suggestions and comments. In the following, we would try to justify the alignment and give more details about the animal experimentation.

The paper lacks clarification on why the normalized mean activity of the tonic stimulation is significantly higher than for biomimetic and mechanical stimulation, as shown in Figure 4. Additionally, it is crucial to detail the methodology used for normalization to ensure transparency and reproducibility. It's important to consider the potential impact of this higher mean activity on other findings and results throughout the study.

We apologize for the lack of clarification and thank the reviewer for pointing it out. We changed the name to “overall normalized activity” as it better describes the methodology. It was used for quantifying the similarity between PSTH values caused by different stimulation patterns, presented on the right part of Figure 4A.

We normalized PSTH values for each condition to be in the range [0,1], by dividing each point of the signal with the maximal signal value (Figure 4A right). With the bars we presented summed activity, divided by the maximum activity among three conditions, in order to make the bars ranging between 0 and 1 (Figure 4A left).

Biomimetically induced activity is much more similar to the amount of activity induced naturally than the one caused by tonic stimulation. We hypothesize that during tonic stimulation the spinal neural networks are overwhelmed with the synchronized information that is constantly received and it could be one of the reasons why paresthesia is often perceived with commonly used neuromodulation paradigms (Tan, D. W. et al. A neural interface provides long-term stable natural touch perception. *Sci Transl Med* 6, 257ra138 (2014); Formento, E., et al. *J Neural Eng* 17, 046019 (2020)). Analogously, the presented similarity of neural natural activity and the one induced with biomimetic stimulation could explain why biomimetic stimulation is perceived as more natural.

We have expanded this section in the manuscript:

“ We normalized the PSTH activity of each condition to be in the range [0,1], by dividing each point of the signal with the maximal signal value (Fig. 4A, right). We summed the activity for each condition and divided it by the maximum activity between three conditions, in order to make the bars ranging between 0 and 1. The bars show the overall normalized activity for the three conditions (Fig. 4A, right). Natural touch and biomimetic stimulation resulted in similar values, while tonic stimulation induced much higher activity in spinal cord and DRG. We hypothesize that during tonic stimulation the spinal neural networks are overwhelmed with synchronized information that is constantly received which causes the paresthesia is often perceived with commonly used neuromodulation paradigms^{37,53}. Analogously, the presented similarity of neural natural activity and the one induced with biomimetic stimulation could explain why biomimetic stimulation is perceived as more natural.”

Including the Current Source Density (CSD) correlation between the tonic and biomimetic stimulation patterns in Figure 4 D, additionally to the shown correlation between each of them and naturalistic mechanical stimulation, would enhance the paper's comprehensiveness and aid in understanding the relationships between these patterns.

We thank the reviewer for the suggestion. We have added the comparison between tonic and biomimetic stimulation neural response in the Figure 4 and expended the manuscript in the Results section: *“We present the CSD estimated using local field potentials induced with biomimetic, tonic electrical stimulation, or natural touch (Fig 4.C). By visually inspecting the spatial distribution of sinks and sources along the spinal axes, and comparing the overall Pearson correlation coefficients between CSDs resulting from different conditions, we can conclude that naturally induced touch response was more similar to the neural signal resulting from biomimetic stimulation (correlation coefficient 0.11, $p=0.005$, $\alpha=0.05$) than to the one produced with constant, 50 Hz electrical stimulation (correlation coefficient $=-0.03$, $p=0.344$, $\alpha=0.05$). As expected, biomimetic and tonic stimulation, as both artificial, electrical stimuli, are also showing CSD similarity (correlation coefficient $=-0.13$, $p=0.001$, $\alpha=0.05$).”*

The choice of using correlation to assess spatio-temporal neural dynamics between DRG and L6 should be elaborated upon in the paper, as it is not sensitive to potential temporal delays. Furthermore, if Peri-Stimulus Time Histograms (PSTH) were employed for correlation analysis (e.g., line 210f), addressing potential effects of bin width on the results is essential to evaluate the robustness of the findings. Overall, this section would benefit from a clarification of the statistical approach.

We thank the reviewer for pointing this out. Together with the results presented in the figure below, we correct and clarify the observations.

When defining the proper size of the time bin, we began by plotting preliminary PSTHs with different time bin lengths and visually inspecting the results. In panel A of the figure we are presenting a neural response to biomimetic stimulation in DRG, using different sizes of the bin. If the bin size is too small (panel A, upper left corner, 1ms bin), the histogram appears to be very noisy, with many bars that are very short and don't provide any clear information about the data's distribution. On contrary, if the bins are too large (panel B, upper right corner, 100ms bin), the histogram lacks resolution, lose important details and do not resemble the neural dynamics well. Therefore, the bins should be large enough to smooth out the noise in the data but not so large that they oversimplify the distribution. In the figure below, panel A down, we are presenting the shape of the PSTH with the optimal bin size. Moreover, we applied Freedman-Diaconis rule for determining the histogram bin width. The optimal bin width is estimated as: $2IQR/N^{1/3}$, where IQR is interquartile range of the data and N is the number of datapoints in our dataset. By applying this transformation, the values varied from 10 to 53ms, depending on the stimulating condition. The results of this estimating approach supported the conclusions of our visual inspection.

Furthermore, we tested the cross-correlation of the PSTHs arising from different stimulation conditions, since it is sensitive to temporal delays (panel B), as suggested by the reviewer. To compute the cross-correlation, one of the signals is kept fixed, while the other signal is shifted in time relative to the reference signal. Each cross-correlation value is associated with a specific time shift (named lag) between the two signals. The lag can be positive, indicating that the reference signal is shifted forward in time compared to the target signal, and vice versa for the negative values. The lag that induces the highest value of the cross-correlation represents the time shift that should be applied in order to align two signals the most. The cross-correlation values resulted when comparing stimulation, DRG and spinal neural response are extremely high. Moreover, the maximum of the cross-correlation is with very small lags (close to 0), which explains that the signals are the most similar with very small time shifts that arise from the short time needed for the signal to be transmitted from the periphery towards the spinal cord. It confirms the hypothesis that biomimetic pattern of activation was transmitted to the DRG and spinal cord maintaining the same spatio-temporal neural dynamics.

We are adding the cross-correlation analysis for two bin sizes (10ms - left, 50ms - right) in order to eliminate the bin width effects on the given conclusions. Even though by using smaller time bins we could see the difference between the maximum lags (which can be even more precise with smaller bin lengths), unveiling the precise latency of the signal when traveling from periphery to DRG and spinal cord was out of the scope of the study.

We have added cross-correlation on the Figure 3, adjusting accordingly the figure caption and expanded the Methods and Results section of manuscript.

In the Methods section we added:

"When defining the proper size of the time bin, we began by plotting preliminary PSTHs with different time bin lengths and visually inspecting the results. If the bin size is too small, the histogram appears to be very noisy, with many bars that are very short and don't provide any clear information about the data's distribution. On the contrary, if the bins are too large, the histogram lacks resolution, lose important details and do not resemble the neural dynamics well. Therefore, the bins should be large enough to smooth out the noise in the data but not so large that they oversimplify the distribution. In the panel A down of the figure we are presenting the shape of the PSTH with the optimal bin size. Moreover, we applied Freedman-Diaconis rule for determining the histogram bin width. The optimal bin width is estimated as: $2IQR/N^{1/3}$, where IQR is interquartile range of the data and N is the number of datapoints in our dataset. By applying this transformation, the values varied from 10 to 53ms, depending on the stimulating condition. The results of this estimating approach supported the conclusions of our visual inspection."

In the Results we added:

" We computed the cross-correlation of the PSTHs derived from the stimulation and neural responses recorded in DRG and spinal cord. The cross-correlation values resulted when comparing stimulation shape, DRG signal and spinal neural response are high. It confirms the hypothesis that biomimetic pattern of activation was transported to the DRG and spinal cord maintaining the same spatio-temporal neural dynamics."

Reporting the stimulation parameters for tibial nerve stimulation in cats, as well as clarifying the definition of 'clear sensory volley', would enhance the method section's clarity and reproducibility.

We thank the reviewer for the suggestion. We stimulated the cat tibial nerve with an 60 μ A amplitude. Of course, this value, is subject- and implant- dependent, and we determinate it with following method. Prior to running the experimental protocol with different stimulations, we were performing tuning trials in which we inspect the effects of the peripheral stimulation to the signal in the spinal cord in the real-time. We were stimulating the nerve pulse by pulse (low frequency stimulation, ~2Hz) and increasing the amplitude until we achieved robust and reproducible afferent volley (clearly observable electrical activity from the spine). These volleys carry the information from the periphery, which is then integrated and processed along the somatosensory neuroaxis. To find this threshold we calculated the mean (meanBaseline) and standard deviation value of the resting signal (stdBaseline): when peak-to-peak values of the afferent volleys were higher than meanBaseline+2.5stdBaseline, we define that amplitude value as the amplitude of stimulation to be used. We are adding this explanation in the "Electrophysiology in decerebrated cats" section of Methods:

"Prior to running the experimental protocol with different stimulations, we were performing tuning trials in which we inspected the effects of the peripheral stimulation to the signal in the spinal cord in the real-time. We stimulated the nerve pulse by pulse (low frequency stimulation, ~2Hz) and increased amplitude until we achieved robust and reproducible afferent volley (clearly observable electrical activity from the spine). These volleys carry the information from the periphery, which is then integrated and processed along the somatosensory neuroaxis. We calculate the mean (meanBaseline) and standard deviation value of the resting signal (stdBaseline). To find this threshold we calculated the mean (meanBaseline) and standard deviation value of the resting signal (stdBaseline): when peak-to-peak values of the afferent volleys were higher than meanBaseline+2.5stdBaseline, we define that amplitude value as the amplitude of stimulation to be used. We stimulated the cat's tibial nerve with 60 μ A amplitude."

The rationale for studying signal propagation in decerebrated cats, which abolishes top-down modulation, while on the other hand top-down modulation might be observed in humans using prosthetic legs, should be explicitly explained in the paper to justify this choice.

Decerebrated cats' experiments allowed us to have an objective insight if the biomimetic paradigms are propagated from the periphery through the somatosensory neuroaxis-which was now known a priori, and compare the pure features of neural response to the natural touch and different stimulation paradigms. Indeed, it is a step prior to understanding the top-down influence also and characterizing the purely sensory component. To do so, we defined a setup that would not disturb the controlled peripheral input and its neural pathway toward the spinal cord, since the contributions of the top-down signals would be extremely difficult to disentangle from signals. Validity of operation of sensorimotor networks obtained in this animal model was demonstrated in studies, in which activity of elements of these networks were recorded also in intact cats during motor behavior (Matsuyama, Drew 2000). Importantly, by using the decerebration method, we also excluded

the necessity to use anesthesia as it can potentially alter the neural signals and eliminate the interference with movements.

We added this point in the discussion:

“Decerebrated cats’ experiments allowed us to objectively inspect the propagation of biomimetic paradigms from the periphery, which was now known a priori, in controlled and undisturbed manner.

This enabled the unique experimental setup for the comparison of the features of neural response to the natural touch and different stimulation paradigms at the spinal level without interfering with descendent signals that would be difficult to disentangle. Using the decerebration method, we also excluded the use of anesthesia as it can alter the neural signals and eliminated the interference with movements.”

The discrepancy between simulated activity (foot sole pressure in humans, duration 2s) and actual mechanical stimulation (cat's leg with a cotton swab, duration 1s) raises concerns about confounding factors in the comparison of correlation patterns. The location of the applied mechanical, naturalistic stimulus using the cotton swab is ill-specified, since only ‘cat’s leg’ is given as a description. Since the biomimetic stimulation is matched to the innervation of the human foot sole, it is not clear how similar this would be to touching arbitrary locations of a cat’s leg. The paper should explicitly acknowledge and discuss how these differences might affect the correlation analysis and the interpretation of results. Demonstrating the equivalence of neural patterns evoked by cotton swab stimulation and stepping is crucial. If available, providing sources that support the similarity in foot and leg innervation between cats and humans would further strengthen the paper's foundation.

We thank the reviewer for this sharp suggestion. During locomotion, activation of sensory fibers transmitting the information from the periphery is similar between cats and humans (Pearcey, Gregory EP, and E. Paul Zehr. "We are upright-walking cats: human limbs as sensory antennae during locomotion." *Physiology* 34.5 (2019): 354-364.), giving us a strong rationale to use cats as the animal models suitable for this study. In the very complex animal experiments we presented (involving premedication, laminectomy, peripheral implants, decerebration, spinal implants, DRG implants, with the tuning of the electrodes positioning, stimulating from peripheral nerve, recording from DRG and spine of decerebrated cat, making the total experimental procedure 12-15 hours long, constantly resolving possible animal health issues) our main goal was not to demonstrate the equivalence of neural patterns evoked by cotton swab stimulation and stepping, but much more basic or rudimental: understanding if the more sophisticated stimulation elicits a more similar cumulative neural signal at the spinal level than the “classic” tonic stimulation. In these pioneering experiments, we were touching latero-caudal leg area, involving the heel, which are innervated by a tibial and common sciatic nerve. These are the parts of the same common nerves going to the paw sole also. By doing so, we activated the afferent types that innervate also human foot sole, as defined biomimetic patterns are developed on the bases of their activity. Our goal was to make a proof of concept of recording of the “gross” signal elicited by touch and electrostimulation, and then analyze its cumulative features, on the population level (Fig.4., Fig. S1). Importantly, we believe that these differences do not affect the main messages:

1. Spatio-temporal features (shape) of the signal -> We conclude that the natural touch neural response follows the shape of variable neural activity, which is stronger at the beginning and end of the stimuli, contrary to the constant tonic stimulation (Fig.4A).

2. The level of neural response change along the transversal spinal axes -> We showed that in the natural touch condition, the signal is not activating different parts of the spinal cord in the same way, i.e. that it is differentially transmitted, more similarly with biomimetic (Fig. S1).

3. The most activated areas in the spinal cord -> We showed that the activated spinal interneurons’ networks are shared between the natural touch and biomimetic more than with tonic stimulation (Fig.4C and D).

4. LFP distribution in DRG. -> We showed a great similarity between natural and biomimetic distributions, while the tonic one is very different. Since we are looking at the overall distribution of the LFP signal resulting from the entire population activity of the DRG (Fig. 4B), it is expected that the LFP distribution we are analyzing would be just slightly changed if the natural touching condition included just other specific foot areas.

The level of equivalence that reviewer is commenting, is commendable and will be absolutely the goal of the next studies, after this one, where we have shown the feasibility of this setup for the first time. Ideally, in the perfect experimental setup, we should apply the mechanical stimuli on the touch cat’s paw in a very controlled way, using a robotic setup that applies the constant, predefined pressure that would elicit robust neural response along the somatosensory neuroaxis. However, that procedure is expensive, complex, and above all very time-consuming, which is of paramount importance in these long terminal animal experiments.

Following the reviewer’s suggestion, we are expanding the Limitation and Discussion section.

We changed the Limitation section, from:

“The natural mechanical stimulation provided in the experimental protocol with decerebrated cats do not exactly replicate the same stimuli. However, the complexity and length of the surgery and experiments were

extremely high, so we applied the stimuli suitable for performing in a reasonable time frame. To do so we delivered rapid tactile stimuli to the cat leg with a cotton swab, in this way we are able to superficially activate multiple sensory fibers in the tibial nerve, increasing therefore the likelihood of effectively eliciting the signals that we can effectively record. Instead, touching spatially distinctive foot sole areas could result in multiple positions not eliciting any activity that we could record, therefore exceeding the available experimental time. Future tests should examine the comparison of PNS with different types of natural tactile stimulation."

To:

"During the animal experiments, ideally, we would apply the mechanical stimuli replicating natural touch on the cat's paw in a controlled way, using a robotic setup that applies a constant, predefined pressure to elicit robust neural response along the somatosensory neuroaxis. However, that procedure is expensive, complex, and above all very time-consuming, therefore was impracticable in our proof-of-concept experiments. Future tests should examine the comparison of PNS with different types of natural tactile stimulation."

We are adding to the Discussion section:

"In this work, we used cats as an animal model due to their similar peripheral sensory fiber activation to humans during locomotion (Pearcey et al. 2019). Decerebrated cats' experiments allowed us to objectively inspect the propagation of biomimetic paradigms from the periphery, which was now known a priori, in controlled and undisturbed manner, giving us a strong rationale to use cats as the animal models suitable for this study." ... In these very complex, 12-15 hours long, animal experiments our main goal was to understand if the more sophisticated stimulation elicits a cumulative neural signal more similar to natural ones, than the "classic" tonic stimulation at the spinal level. In these pioneering experiments, we were touching latero-caudal leg area, involving the heel, which are innervated by a tibial and common sciatic nerve. These are the parts of the same common nerves going to the paw sole also. By doing so, we activated the afferent types that innervate also human foot sole, as defined biomimetic patterns are developed on the bases of their activity. Our goal was to make a proof of concept of recording of the "gross" signal elicited by touch and electrostimulation, and then analyze its cumulative features, on the population level."

The decision to fix stimulation amplitude beforehand and not modulate it within the stimulation, despite previous publications suggesting its potential benefits (in hand prosthetics, Valle et al., 2018), should be discussed.

We apologize for this misunderstanding. It is true that for naturalness assessment, we compared only frequency modulated encodings, since previous studies have shown that frequency is the stimulation parameter strictly linked to evoked sensation quality (Valle et al., 2018 Neuron, Graczyk et al., 2021 J Neurosci, Hughes et al., 2021 eLife) while amplitude seems to modulate less sensation naturalness (Valle et al., 2018 Neuron) and more sensation intensity. This choice was also made to avoid too long stimulation sessions (too many conditions) that could result in adaptation and bias in perceived quality. We added this information in the methods:

"Amplitude and pulse-width of stimulation, identified during the electrode mapping procedure, were kept constant along the train. This choice was based on previous findings that frequency is the stimulation parameter most linked to evoked sensation quality (Valle et al., 2018 Neuron, Graczyk et al., 2021 J Neurosci, Hughes et al., 2021 eLife) while charge seems to influence it less (Valle et al., 2018 Neuron)."

Importantly, since the injected charge is modulated to encode sensation intensity (Valle et al., 2018 Sci Rep), we included this modulation during the walking experiments with the neuroprosthetic legs resulting in an encoding based on co-modulation of frequency and charge. Indeed, in the methods, we have included: "The amplitude of the stimulation was modulated linearly with the pressure sensor output, as proposed in Valle et al., (HNM-1)⁴¹"

It is worth noting that the animal experiment presented an opportunity for fine-tuning the stimulation parameters to potentially achieve a more robust biomimetic activation in the spinal cord neurons. However, this valuable adjustment remained unexplored, and its absence from the report raises questions about the optimization of experimental outcomes. Providing insights into why this fine-tuning was not pursued would enhance the transparency of the research process. Detailing the exact purpose of the animal experiment and discussing how different outcomes could have influenced the experimental protocol for the patient study is important for transparency and understanding the study's design, as well as for guiding future research.

Prior to this study, we assumed, but we did not know, if the biomimetic stimulations we designed are transmitted at all through the somatosensory neuroaxis, and with any difference w.r.t. "classical paresthesia-like" tonic stimulation. Therefore, this was the main purpose of the animal experiments. Moreover, it was not known how the frequency modulation affects neural response and we were able to understand the neural dynamics' changes in the DRG and spinal cord. If the outcome was to be that tonic stimulation had the same results as biomimetics patterns, then it would not make sense to have it in experimental protocol for the patient study.

Moreover, these extremely complex experiments (15 h overall) after surgeries that lasted between 9 and 11 hours involved a lot of manual fine-tuning of stimulation and ad-hoc understanding performed in the overall 12-15h of animal testing. When, after many hours, we achieved the reliable stim (eliciting visible spinal volleys, abovementioned and described in the Methods), we could go with the experiments further. The time of animal's survival was therefore the limiting factor in further optimizing huge parameter space. However, finally, as animals cannot report the perceived sensation, we applied the approach in humans and tested the hypothesis that the stimulation paradigm that induces a more similar neural response as natural touch is also perceived more natural and increases the performance of the prosthetic use.

Reviewer #2

This manuscript describes a multi-faceted approach to the design, implementation, and evaluation of a biomimetic approach to restoring touch sensation via implanted peripheral nerve interfaces. The study is well designed and includes computational modelling, animal experiments, and trials in humans. The potential benefits of biomimetic stimulation are well known and widely accepted, so the main contribution of this work is the combined multi-faceted approach and the results of such as applied to the lower limb. The methodology is sound, although there are several areas of confusion in the data analysis that require clarification.

We thank the reviewer for the introductory comment and the appreciation of our study, especially for acknowledging the importance of a multifaceted approach.

Major comments

Although the content of the manuscript is clear, the quality of the writing requires improvement. I suggest using a proofing service to revise the manuscript. There are minor grammatical mistakes in most sentences, and this significantly detracts from the scientific quality.

We thank the reviewer for his/her suggestion. We performed the proof-reading by a native English corrector.

On line 318 it is clearly stated that three human participants were recruited, however line 342 refers to "both" subjects and all results from this point on only cover two participants. I could not find the rationale for only presenting data from two out of three participants - all results need to be included. Likewise, I was confused if only data from Cat 1 is presented in Figure 4 a, and if so why?

We apologize for the lack of clarity. Three transfemoral amputees were recruited and implanted in their leg nerves. While Subjects 1-2 participated in all experiments, because of limited time availability, Subject 3 participated only in open-loop, characterization and the assessment of naturalness and decided not to participate in these functional tasks (ST and CDT). Data from the open-loop sessions testing the naturalness of the evoked sensations for Subject 3 is reported in Fig. S2. We added more information about this point in the Methods section:

"This study was performed within a larger set of experimental protocols aimed at assessing the impact of the restoration of sensory feedback via neural implants in three leg amputees^{7,36,49,50,85}. The data reported in this manuscript was obtained in multiple days from randomized conditions. The functional data was acquired within 2-weeks to minimize possible training effects, which are probably to exclude (Fig. S5). While Subjects 1-2 participated in all experiments, because of limited time availability, Subject 3 participated only in the open-loop characterization and the assessment of naturalness while deciding not to participate in these functional tasks with biomimetics (ST and CDT). Data from the open-loop sessions testing the naturalness of the evoked sensations for Subject 3 is reported in Fig. S2."

Regarding animal experiments, the decerebration procedure in cats is a highly delicate and invasive surgical technique (Whelan, Patrick J. "Control of locomotion in the decerebrate cat." *Progress in neurobiology* 49.5 (1996): 481-515.). It is followed with risks that include bleeding, infection, tissue damage, complications related to surgical techniques, as well as problems with the respiration. These extremely complex experiments (premedication, laminectomy, peripheral implants, decerebration (Merkulyeva et.al 2018), spinal implants, DRG implants, with the preparing the electrodes positioning, lasted between 9 and 11 hours, making the total experimental procedure 12-15 hours long) involving a lot of manual fine-tuning. While with the first cat we managed to perform the whole protocol, the issue with the animal health stability occurred during the second animal experiments and we managed to perform only the natural touch condition, and unfortunately couldn't apply any electrical stimuli after that moment. However, experiments with the cat 2 helped us to understand that natural touch recorded neural response is not exclusively subject specific, but that the neural dynamics signals, recorded for the first time, follow a similar trend between different cats.

Minor comments

It would be good to see more discussion about the differences between upper and lower limb amputees; they have very different needs in terms of sensory feedback and this needs be discussed in order to place this work in the context of existing literature on upper limb amputees.

As suggested by the reviewer, we added a paragraph in the Discussion on this specific point and more reference from studies in upper-limb amputees:

"In addition to our results, previous studies have shown preliminarily improvements in manual dexterity and object recognition in upper-limb amputees via robotic hand prostheses^{40,41}. Nevertheless, there are noticeable differences to consider between upper and lower limb prosthetics in the design of synthetic sensory feedback. First, the sciatic nerve, which innervates the foot and lower leg, is more than twice as large as the median and ulnar nerves, which innervate the fingers and palms, and it is also difficult to reach through the big leg muscles during the surgery. The density and placement of the receptors are different between upper and lower extremities (considered in TouchSim and FootSim). Upper-limb amputees can use their intact hand for almost all activities, while leg amputees cannot ambulate without a prosthesis. A failed manipulation can lead to a broken glass, whereas a failed step could lead to a dangerous fall. Finally, more proximal levels of amputation require more complexity in the sensory feedback restoration, as larger amounts of information need to be transmitted. These feedback specifications need to be specifically considered for the different amputation types."

Importantly, previous studies on upper-limb amputees have then shown that sensory feedback increased subjects' confidence (i.e., self-efficacy) and was directly correlated with prosthesis accuracy in functional tasks (Schiefer et al., 2018 PlosOne, Graczyk et al., 2019 PlosOne). In that case the increments between the use of the prosthesis with and without tactile feedback were between +40% and +60% in object identification. In our case, the improvements of using the prosthesis with and without biomimetic feedback were similar +46% and +58% for S1 and S2 respectively. As suggested, to contextualize our results, we included in the discussion:

"Moreover, previous studies on upper-limb amputees have then shown that sensory feedback increased subjects' confidence (i.e., self-efficacy) and was directly correlated with prosthesis accuracy in functional tasks (Schiefer et al., 2018 PlosOne, Graczyk et al., 2019 PlosOne)."

Regarding the naturalness of the direct neural stimulation, low values (around 2/5) were also reported in upper-limb amputees when using linear and non-biomimetic approaches (Valle et al., 2018 Neuron).

In Fig 2 I am not clear how the FULL stimulation pattern was derived, visually it does not appear to be the sum of the others?

We thank the reviewer for pointing this out, in the Figure 2 there was a mistake in the presented FA2 pattern. We corrected the pattern and corresponding layout and slightly polished the figure (linewidth, positioning etc.). All stimulation patterns were generated using FootSim, populating the foot sole with a specific fiber type or with all types of afferents. The FULL biomimetic pattern considers the complete foot sole population and its activity. Activation of each single fiber is superimposed to create a combined response. The spiking activity is aggregated in the form of the peri-stimulus time histogram, the function is smoothed and used as a modulated frequency of stimulation. The procedure is the same for each biomimetic stimulation pattern. Considering these steps, the FULL biomimetic paradigm represents the summarized afferent activity, however it does not correspond to the exact summation of stimuli spikes from all the other biomimetic patterns.

We have added more precise explanation in the Methods of the manuscript:

"The FootSim model estimates the response of each single afferent placed on the sole of the foot and provides as its spiking activity. This activity for all fibers placed in the foot sole is aggregated using peri-stimulus time histogram (PSTH). We fit the smooth function resembling the PSTH and use it to modulate the frequency of stimulation. We applied the same procedure for each biomimetic paradigm. Amplitude and pulse-width of stimulation, identified during the electrode mapping procedure, were kept constant along the train. This choice was based on previous finding that frequency is the stimulation parameter more linked to evoked sensation quality^{41,61,90} while charge seems to modulate intensity⁹¹."

When the cuff electrodes were implanted in the cats the manuscript states (line 197) that stimulation was increased to be slightly above threshold - what threshold?

We thank the reviewer for pointing out this misunderstanding. We are adding this explanation in the "Electrophysiology in decerebrated cats" section of Methods:

"Prior to running the experimental protocol with different stimulations, we were performing tuning trials in which we inspected the effects of the peripheral stimulation to the signal in the spinal cord in the real-time. We stimulated the nerve pulse by pulse (low frequency stimulation, ~2Hz) and increased amplitude until we achieved robust and reproducible afferent volley (clearly observable electrical activity from the spine). These volleys carry the information from the periphery, which is then integrated and processed along the somatosensory neuroaxis. We calculate the mean (meanBaseline) and standard deviation value of the resting signal (stdBaseline). To find

this threshold we calculated the mean (meanBaseline) and standard deviation value of the resting signal (stdBaseline): when peak-to-peak values of the afferent volleys were higher than meanBaseline+2.5stdBaseline, we define that amplitude value as the amplitude of stimulation to be used. We stimulated the cat's tibial nerve with 60 μ A amplitude."

What is the rationale for using cuffs in cats and TIMEs in humans? I would expect the spatial selectivity of TIMEs to be much higher.

We thank the reviewer for this sharp observation. The purpose of using a cuff electrode instead of TIME relates to: 1) the need of an electrode for stimulating the whole nerve trunk and not subregions of the target nerve, since for the experiment highly selectivity was not required-we wanted to robustly induce and study the neural activity at the spine level 2) given the huge complexity of the recording setup in cats (premedication, laminectomy, peripheral implants, decerebration, spinal implants, DRG implants, pre-tuning of the electrodes stimulation), we adopted an easier to implant electrode, so to be able to perform experiments during the health-window of animal (during the long experimental duration (15h)). Using TIMEs would require hours more of fine-tuning of active sites eliciting volleys, while indeed we were interested in the evoked neural dynamics more than spatial selectivity to understand the information propagation over the neuroaxis.

In Fig 3 c), are the orange waveforms the mean event rate?

We apologize for the mistake, mean event rate explanation was accidentally left from our previous way of quantifying the PSTH that we abandoned in this version of the manuscript.

To clarify, orange/yellow waveforms present the peri-stimulus time histogram (PSTH). PSTHs are histograms used to visualize the rate and timing of neuronal spike discharges in relation to an external stimulus or event.

In all figures the labels need to be improved. Please make the letters (a, b) larger and consider more consistent use of boxes to show which parts of the figure the labels apply to.

We thank the reviewer for this suggestion. We tried to correct it accordingly in all the figures.

When the cats' paws were stimulated with a cotton bud how did you synchronize this with the neural recording? Fig 4 a) implies a mechanical stimulus that was acting down on the paw and not a cotton bud. Why is the spiking activity shown in respect of "stimuli length" and not time?

We thank the reviewer for pointing out this misunderstanding. During the experimental procedure, we had a trigger (rope connected with encoder on one side and animal's tail on the other) that was pulled with every stimulation of cat's leg as well as video recordings that we used for the synchronization of the trial, both were recorded with neural data as separate channels. Between every natural touch stimuli, the pause was made in order to ensure the ability to extract the data with no stimulation. Moreover, when extracting the data of interest during natural touch conditions, we looked at the neural spiking data and extracted the time period in which we saw the single unit activity, which corresponds to the period that can be extracted using trigger and video recordings. We have now changed the "stimuli length" to the time in seconds.

In the human trials, the order of the tasks was randomised. However, there will still have been some learning as the participants repeated the task - how did you account for this?

We thank the reviewer for this comment. First, we added a new Supplementary figure (Fig. S5) showing that there is no effect in learning or habituation in our Dual Task paradigm.

Fig S5

Both subjects showed low Spearman's rank correlation coefficients (between 0.03 and 0.5) across sessions, suggesting almost no learning or initial habituation effect on their spelling performance. In the results section, we added:

"Analyzing the spelling accuracy, both subjects showed low Spearman's rank correlation coefficients (between 0.03 and 0.5) across sessions, suggesting no learning or initial habituation effect on their performance (**Fig.S5**)."

Since learning is a phenomenon that can bias results, we avoid it few different manners: 1) we randomized all the conditions in order to avoid biases in a specific stimulation paradigm; 2) we run all the functional tests in the same week to avoid effect of long-term training.

We added this information in the method section: "The data reported in this manuscript was obtained in multiple days with randomized conditions. *The functional data was acquired within a same week to minimize possible training effects, which are probably to exclude (see Fig. S5).*"

I was surprised that the self-reported confidence did not show as significant an improvement as the speed of the task. Do you know why this was?

The self-reported confidence showed significant improvements as the speed results, always in the same direction a part in Subject 2 LIN and NF VS DISC:

"Interestingly, also the self-reported confidence (VAS scale 0-10) in walking on stairs was increased, when the participants were exploiting the neuroprosthetic device with biomimetic neurofeedback (9.75 ± 0.26 for S1 and 6 ± 0.3 for S2) compared to LIN (8.75 ± 0.62 , $p < 0.05$ for S1 and 5.37 ± 0.23 , $p < 0.05$ for S2), DISC (7.83 ± 0.39 , $p < 0.05$ for S1 and 5.17 ± 0.25 , $p < 0.05$ for S2) and NF (6.67 ± 0.49 , $p < 0.05$ for S1 and 3.83 ± 0.25 , $p < 0.05$ for S2) conditions"

Our working hypothesis is that confidence and mobility result to be among the clearest and simplest parameters showing the impact of sensory feedback on gait. Therefore, we suggest considering these important features as the global evidence of a better sensorimotor strategy adopted by the subject. This could be of great importance in the design and the evaluation of the benefits of new somatosensory neuroprostheses. However, their direct relationship has not been quantified yet in order to directly link specific improvements on self-confidence to walking speed.

Many studies have reported the decreased self-confidence in walking for lower-limb amputees and its strong relationship with mobility and walking abilities (Miller et al., 2001 APMR, Miller et al., 2011 Prosthetics and Orthotics International and Sions JM, et al., 2020 Physiother Theory Pract.). Notably, previous studies on upper-limb amputees have then shown that sensory feedback increased subjects' confidence (i.e., self-efficacy) and was directly correlated with prosthesis accuracy in functional tasks (Schiefer et al., 2018 PlosOne, Graczyk et al., 2019 PlosOne). In that case the increments between the use of the prosthesis with and without tactile feedback

were between +40% and +60% in object identification. In our case, the improvements of using the prosthesis with and without biomimetic feedback were similar +46% and +58% for S1 and S2 respectively.

Lines 486/487, please do not use "him" when referring to participants unless you are specifically referring to a male participant. Please use "them/they" instead.

We apologize for this mistake. We changed the term in the manuscript to "the subjects" or to "they/them".

REVIEWERS' COMMENTS

Reviewer #2 (Remarks to the Author):

The authors have satisfactorily addressed my concerns and have made helpful changes to the manuscript. The revised figures and text are welcome and I have no further comments to be addressed.

Reviewer #2 commenting on Reviewer #1's concerns raised in the previous round of review:

The authors have responded in detail to each point (sometimes with the same answer they gave to me on similar points), and I feel that they have suitably improved the paper.